# Evolution of the hypoxia-sensitive cells involved in amniote respiratory reflexes

**Dorit Hockman[1,2,3], Alan J Burns[4,5†], Gerhard Schlosser[6], Keith P Gates[7], Benjamin Jevans[4], Alessandro Mongera[8‡], Shannon Fisher[9§], Gokhan Unlu[10], Ela W Knapik[10], Charles K Kaufman[11¶**††], Christian Mosimann[11‡‡], Leonard I Zon[11], Joseph J Lancman[7], P Duc S Dong[7], Heiko Lickert[12], Abigail S Tucker[13], Clare V H Baker[1*]**

[1]Department of Physiology, Development and Neuroscience, University of Cambridge, Cambridge, United Kingdom; [2]Weatherall Institute of Molecular Medicine, University of Oxford, Oxford, United Kingdom; [3]Department of Molecular and Cell Biology, University of Cape Town, Cape Town, South Africa; [4]Stem Cells and Regenerative Medicine, UCL Great Ormond Street Institute of Child Health, London, United Kingdom; [5]Department of Clinical Genetics, Erasmus Medical Center, Rotterdam, The Netherlands; [6]School of Natural Sciences, National University of Ireland, Galway, Ireland; [7]Human Genetics Program, Sanford Burnham Prebys Medical Discovery Institute, La Jolla, United States; [8]Department of Genetics, Max-Planck Institut für Entwicklungsbiologie, Tübingen, Germany; [9]Department of Cell and Developmental Biology, University of Pennsylvania, Philadelphia, United States; [10]Division of Genetic Medicine, Department of Medicine, Vanderbilt University Medical Center, Nashville, United States; [11]Children's Hospital Boston, Howard Hughes Medical Institute, Harvard Medical School, Boston, United States; [12]Institute of Diabetes and Regeneration Research, Helmholtz Zentrum München, Neuherberg, Germany; [13]Department of Craniofacial Development and Stem Cell Biology, King's College London, London, United Kingdom

*For correspondence: cvhb1@cam.ac.uk

Present address: [†]Gastrointestinal Drug Discovery Unit, Takeda Pharmaceuticals United States, Inc., Cambridge, United States; [‡]Department of Mechanical Engineering, California NanoSystem Institute, University of California, Santa Barbara, Santa Barbara, United States; [§]Department of Pharmacology & Experimental Therapeutics, Boston University School of Medicine, Boston, United States; [¶] Department of Medicine, Washington University School of Medicine, St. Louis, United States; [**]Division of Oncology, Washington University School of Medicine, St. Louis, United States; [††]Department of Developmental Biology, Washington University School of Medicine, St. Louis, United States; [‡‡]Institute of Molecular Life Sciences, University of Zürich, Zürich, Switzerland

**Competing interests:** The authors declare that no competing interests exist.

**Abstract** The evolutionary origins of the hypoxia-sensitive cells that trigger amniote respiratory reflexes – carotid body glomus cells, and 'pulmonary neuroendocrine cells' (PNECs) - are obscure. Homology has been proposed between glomus cells, which are neural crest-derived, and the hypoxia-sensitive 'neuroepithelial cells' (NECs) of fish gills, whose embryonic origin is unknown. NECs have also been likened to PNECs, which differentiate in situ within lung airway epithelia. Using genetic lineage-tracing and neural crest-deficient mutants in zebrafish, and physical fate-mapping in frog and lamprey, we find that NECs are not neural crest-derived, but endoderm-derived, like PNECs, whose endodermal origin we confirm. We discover neural crest-derived catecholaminergic cells associated with zebrafish pharyngeal arch blood vessels, and propose a new model for amniote hypoxia-sensitive cell evolution: endoderm-derived NECs were retained as PNECs, while the carotid body evolved via the aggregation of neural crest-derived catecholaminergic (chromaffin) cells already associated with blood vessels in anamniote pharyngeal arches.

**eLife digest** The carotid bodies are small glands found in either side of our neck, near the carotid artery. When the level of oxygen in our blood drops, specialized cells in the carotid bodies signal to the brain to increase our heart rate and make us breathe more rapidly and deeply. As a result, more oxygen is delivered to our cells.

Fish have similar oxygen-sensitive cells in their gills, known as neuroepithelial cells, that detect changes in the oxygen levels in the surrounding water and their blood. It has been suggested that after our vertebrate (back-boned animal) ancestors moved onto land, the neuroepithelial cells in their gills eventually evolved to form the carotid bodies. Knowing whether this is true would allow researchers to better understand how our ancestors were able to adapt to an obligate air-breathing lifestyle on land.

If the carotid body did evolve from ancestral neuroepithelial cells, we would expect that they would both develop from the same kind of cells in the embryo. Carotid body cells develop from a group of cells called neural crest cells, which give rise to many tissues, including nerve cells. Hockman et al. have now investigated whether neuroepithelial cells also develop from neural crest cells.

Hockman et al. labelled the neural crest cells in the embryos of zebrafish, frogs and lampreys using techniques such as injecting the cells with fluorescent dye or genetically modifying the cells to make fluorescent proteins. Unexpectedly, the neuroepithelial cells that developed in the gills of these embryos did not contain these fluorescent labels, meaning that they did not develop from the neural crest cells. The patterns of gene activity found in the developing neuroepithelial cells were also different from those in the carotid body. Further investigation revealed that neuroepithelial cells develop from the lining of the mouth and gills and may be related to a similar population of oxygen-sensitive cells found in the lungs.

Overall, it appears that the carotid body did not evolve from ancestral neuroepithelial cells. However, Hockman et al. did find some cells near blood vessels in the gills of zebrafish that had developed from neural crest cells. Equivalent cells in our ancestors could therefore be the cells that evolved into carotid bodies. A first test of this theory will be to determine whether or not these cells are oxygen-sensitive.

## Introduction

During hypoxia in vertebrates, respiratory reflexes such as hyperventilation are triggered by neurotransmitter release from hypoxia-sensitive serotonergic cells associated with pharyngeal arch arteries, as well as in the lungs and/or gills (reviewed by *López-Barneo et al., 2016*; *Cutz et al., 2013*; *Jonz et al., 2016*). In amniotes, these are the 'glomus cells' of the carotid body, located at the bifurcation of the common carotid artery (reviewed by *Nurse, 2014*; *López-Barneo et al., 2016*), and the 'pulmonary neuroendocrine cells' (PNECs) of lung airway epithelia, found either as solitary, flask-shaped cells, or collected into 'neuroepithelial bodies' (NEBs), preferentially located at airway branch points (reviewed by *Cutz et al., 2013*). Glomus cells respond to hypoxia in arterial blood by releasing stored neurotransmitters including acetylcholine, ATP, the catecholamine dopamine, and serotonin (the latter two most likely acting as autocrine/paracrine neuromodulators) (reviewed by *Nurse and Piskuric, 2013*; *Nurse, 2014*). These excite afferent terminals of the carotid sinus nerve (a branch of the glossopharyngeal nerve, arising from neurons in the petrosal ganglion) in mammals (reviewed by *Nurse and Piskuric, 2013*; *Nurse, 2014*), and of the vagal nerve (arising from neurons in the nodose ganglion) in birds (*Kameda, 2002*). The afferent nerves relay signals to the nucleus of the solitary tract within the hindbrain, to elicit respiratory reflex responses such as hyperventilation (reviewed by *Teppema and Dahan, 2010*). PNECs respond to hypoxia by releasing stored serotonin and various neuropeptides onto vagal afferents, and are thought to act as hypoxia-sensitive airway sensors (reviewed by *Cutz et al., 2013*; also see *Branchfield et al., 2016*). PNECs also provide an important stem-cell niche for regenerating the airway epithelium after injury (*Reynolds et al., 2000*; *Guha et al., 2012*; *Song et al., 2012*) and were recently shown to be the predominant cells of origin for small cell lung cancer (*Park et al., 2011*; *Sutherland et al., 2011*; *Song et al., 2012*).

The evolution of glomus cells was critical for the transition from aquatic life, where externally facing hypoxia-sensors are essential for monitoring the variable oxygen levels in water, to fully terrestrial life, where reflex responses to variations in internal oxygen levels are more important, given the stability of oxygen levels in air (*Burleson and Milsom, 2003*; *Milsom and Burleson, 2007*). However, the evolutionary history of glomus cells - and, indeed, PNECs - is uncertain. One commonly suggested hypothesis (e.g., *Milsom and Burleson, 2007*; *Hempleman and Warburton, 2013*; *Jonz et al., 2016*) is that glomus cells are homologous to the chemosensory 'neuroepithelial cells' (NECs) of fish gills. These were originally identified within the primary epithelium of the gills in various teleosts and a shark, as innervated cells (isolated or clustered) containing dense-cored vesicles; formaldehyde-induced fluorescence revealed the presence of biogenic amines, identified as serotonin, while electron microscopy following exposure of trout to acute hypoxia revealed fewer vesicles, which appeared degranulated (*Dunel-Erb et al., 1982*). In vitro patch-clamp studies on NECs isolated from zebrafish and catfish gills (identified by neutral red, a vital dye that stains monoamine-containing cells including serotonergic PNECs; *Youngson et al., 1993*), confirmed that teleost gill NECs are hypoxia-sensitive (*Jonz et al., 2004*; *Burleson et al., 2006*). The conventional marker for teleost gill NECs is serotonin: although non-serotonergic gill NECs were identified in adult zebrafish by immunoreactivity for synaptic vesicle glycoprotein 2, these may represent immature NECs (*Jonz and Nurse, 2003*; *Jonz et al., 2004*). NECs are found near efferent gill arteries and on the basal lamina of the gill epithelia, facing the flow of water, hence can detect hypoxia and other stimuli in either blood or external water (reviewed by *Jonz et al., 2016*). Like glomus cells (reviewed by *Nurse, 2014*; *López-Barneo et al., 2016*), zebrafish gill NECs also respond to acid hypercapnia (increased $CO_2/H^+$) (*López-Barneo et al., 1988*; *Buckler, 1997*; *Jonz et al., 2004*; *Qin et al., 2010*). The putative shared evolutionary ancestry of glomus cells and gill NECs (e.g. *Milsom and Burleson, 2007*; *Hempleman and Warburton, 2013*; *Jonz et al., 2016*) is supported by many similarities: both are associated with pharyngeal arch arteries (the carotid body develops in association with the third pharyngeal arch artery, which will form the carotid artery), provided with afferent innervation by glossopharyngeal and/or vagal nerves, and have background ('leak') $K^+$ currents that are inhibited by hypoxia, resulting in membrane depolarization, activation of voltage-gated $Ca^{2+}$ channels, and neurotransmitter release (*López-Barneo et al., 1988*; *Buckler, 1997*; *Jonz et al., 2004*; *Qin et al., 2010*).

If glomus cells and gill NECs evolved from the same ancestral cell population, they should share a common embryonic origin. Glomus cells are neural crest-derived, as demonstrated in birds by quail-chick neural fold grafts (*Le Douarin et al., 1972*; *Pearse et al., 1973*), and in mouse by *Wnt1-Cre* genetic lineage-tracing (*Pardal et al., 2007*), but the embryonic origin of NECs is unknown. Lack of immunoreactivity for the HNK1 antibody, which labels migrating neural crest cells in many but not all vertebrates, has been reported for NECs (*Porteus et al., 2013*, *2014*). However, the carbohydrate epitope recognized by the HNK1 antibody (*Voshol et al., 1996*) is borne by multiple glycoproteins and glycolipids, and gene or antigen expression in itself cannot indicate lineage. An alternative to the hypothesis that gill NECs and glomus cells evolved from a common ancestral cell population is that NECs share ancestry with PNECs, to which they were originally likened (*Dunel-Erb et al., 1982*). Hypoxia-sensing by PNECs, as in NECs and glomus cells, involves inhibition of a $K^+$ current by hypoxia (reviewed by *Cutz et al., 2013*; *Nurse, 2014*; *López-Barneo et al., 2016*; *Jonz et al., 2016*). In contrast to the neural crest origin of glomus cells (*Le Douarin et al., 1972*; *Pearse et al., 1973*; *Pardal et al., 2007*), PNECs have an intrinsic pulmonary epithelial origin: the first experimental support for this was provided by a tritiated thymidine labeling study of hamster lung development (*Hoyt et al., 1990*), later confirmed by mouse genetic lineage-tracing studies using *Id2-CreER^{T2}*, *Shh-Cre*, *Nkx2.1-Cre*, and *Sox9-Cre* driver lines that showed a common origin for all lung airway epithelial cell types, including PNECs (*Rawlins et al., 2009*; *Song et al., 2012*; *Kuo and Krasnow, 2015*).

Here, we demonstrate that neural crest cells do not contribute to gill NECs: instead, these are endoderm-derived. This refutes the hypothesis that glomus cells and gill NECs evolved from a common ancestral cell population, and instead supports an evolutionary relationship between NECs and PNECs, whose endodermal origin we confirm in mouse. We also show that the transcription factor Phox2b, which is required for glomus cell development (*Dauger et al., 2003*), is not expressed by gill NECs (or by PNECs), arguing against the possibility of cell-type homology between glomus cells and gill NECs via activation of the same genetic network. Finally, we report the discovery of neural

crest-derived chromaffin (catecholaminergic) cells associated with blood vessels in the pharyngeal arches of juvenile zebrafish, which we speculate could share an evolutionary ancestry with glomus cells. Given these results, we propose a new model for the evolution of hypoxia-sensitive cells during the transition to terrestrial life.

## Results

Carotid body glomus cells develop from the neural crest (*Le Douarin et al., 1972*; *Pearse et al., 1973*; *Pardal et al., 2007*), while PNECs differentiate in situ within pulmonary airway epithelia (*Hoyt et al., 1990*; *Rawlins et al., 2009*; *Song et al., 2012*; *Kuo and Krasnow, 2015*). We aimed to shed light on the evolutionary origins of these amniote hypoxia-sensitive cell types by determining the embryonic origin of NECs, the hypoxia-sensitive cells of anamniote gills.

### Zebrafish NECs are not neural crest-derived

In developing zebrafish, gill NECs were previously identified as serotonin (5-HT)-immunoreactive cells in gill filaments from 5-dpf, which are innervated by 7-dpf (*Jonz and Nurse, 2005*). We investigated any neural crest contribution to gill NECs via genetic lineage-tracing, using a collection of transgenic zebrafish lines with different *cis*-regulatory sequences driving Cre and subsequent lineage reporter expression in neural crest-derived cells, as well as via neural crest-deficient zebrafish embryos. Lineage-tracing using different Cre driver and reporter lines enabled us to control for false negatives potentially arising from variable promoter activity in, or incomplete labeling of, some neural crest cells in either neural crest driver or reporter lines.

In larval and juvenile zebrafish, we identified NECs in the gill filaments by serotonin immunoreactivity, as previously reported (*Jonz and Nurse, 2005*); we also found similar innervated serotonergic cells scattered in the orobranchial epithelium (putative NECs). NECs in the gill filaments, and putative NECs in the orobranchial epithelium, were unlabeled in larvae/metamorphic juveniles with *sox10 cis*-regulatory sequences driving Cre and resulting lineage reporter expression [*Tg(-28.5sox10:cre); Tg(ef1a:loxP-DsRed-loxP-EGFP)* (*Kague et al., 2012*); *Tg(-4.9sox10:creER^{T2});Tg(βactin:loxP-Super-Stop-loxP-DsRed)* (*Mongera et al., 2013*)], even when nearby neural crest derivatives such as branchial arch cartilages/mesenchyme and/or gill pillar cells (*Mongera et al., 2013*) were reporter-positive (*Figure 1a–f'*; 183 serotonergic cells located near such reporter-positive cells [≥6 per fish] were counted across 11 larvae/metamorphic juveniles: n = 8 for −28.5sox10; n = 3 for −4.9sox10). Gill NECs, and putative NECs in the orobranchial epithelium, were also unlabeled in *Tg(crestin:creER^{T2});Tg(-3.5ubi:loxP-GFP-loxP-mCherry)* (*Mosimann et al., 2011*; *Kaufman et al., 2016*) larvae/metamorphic juveniles, even when nearby neural crest-derived cells in branchial arch cartilages and/or gill filament mesenchyme were mCherry-positive (*Figure 1g-h'*; 166 serotonergic cells located near such mCherry-positive cells [≥6 per fish] were counted across 10 larvae/metamorphic juveniles). We ruled out the possibility that lack of lineage reporter expression in NECs was a false-negative result arising from inactivity in NECs of the promoters driving the Cre-switchable reporter cassettes, by confirming that NECs expressed the native unrecombined reporter in both transgenic lines [*Tg(-28.5sox10:cre);Tg(ef1a:loxP-DsRed-loxP-EGFP)* and *Tg(crestin:creER^{T2});Tg(-3.5ubi:loxP-GFP-loxP-mCherry)*] (*Figure 1—figure supplement 1*).

Finally, we analyzed *tfap2a^{mob};foxd3^{mos}* zebrafish, which lack neural crest derivatives (*Barrallo-Gimeno et al., 2004*; *Montero-Balaguer et al., 2006*; *Wang et al., 2011*). At 7-dpf, *tfap2a^{mob}; foxd3^{mos}* mutants lacked pigment cells and lower jaw structures (n = 8; *Figure 2a*). In the absence of the neural crest-derived pharyngeal endoskeleton, pharyngeal arches and gills are hard to recognize, but putative NECs, visualized here as serotonergic cells associated with HNK1 epitope-immunoreactive neurites (*Metcalfe et al., 1990*), were still present in the orobranchial epithelium (*Figure 2b,c*) (and occasionally could also be identified ventral to the orobranchial cavity, where the pharyngeal arches would be located; *Figure 2d,e*). We counted all putative NECs in the orobranchial region of three *tfap2a^{mob};foxd3^{mos}* larvae (294 putative NECs counted in total) and three wild-type siblings (244 putative NECs counted in total): there was no change in mean number (mean/embryo ± s.d.: 81.3 ± 24.0 for wild-type larvae, n = 3; 98.0 ± 49.7 for *tfap2a^{mob};foxd3^{mos}* larvae, n = 3; p=0.63, unpaired two-tailed Student's t-test) (*Figure 2f–i*).

Putative NECs have also been identified as serotonergic cells in the skin of embryonic zebrafish (*Jonz and Nurse, 2006*; *Coccimiglio and Jonz, 2012*) and of adult mangrove killifish, which respire

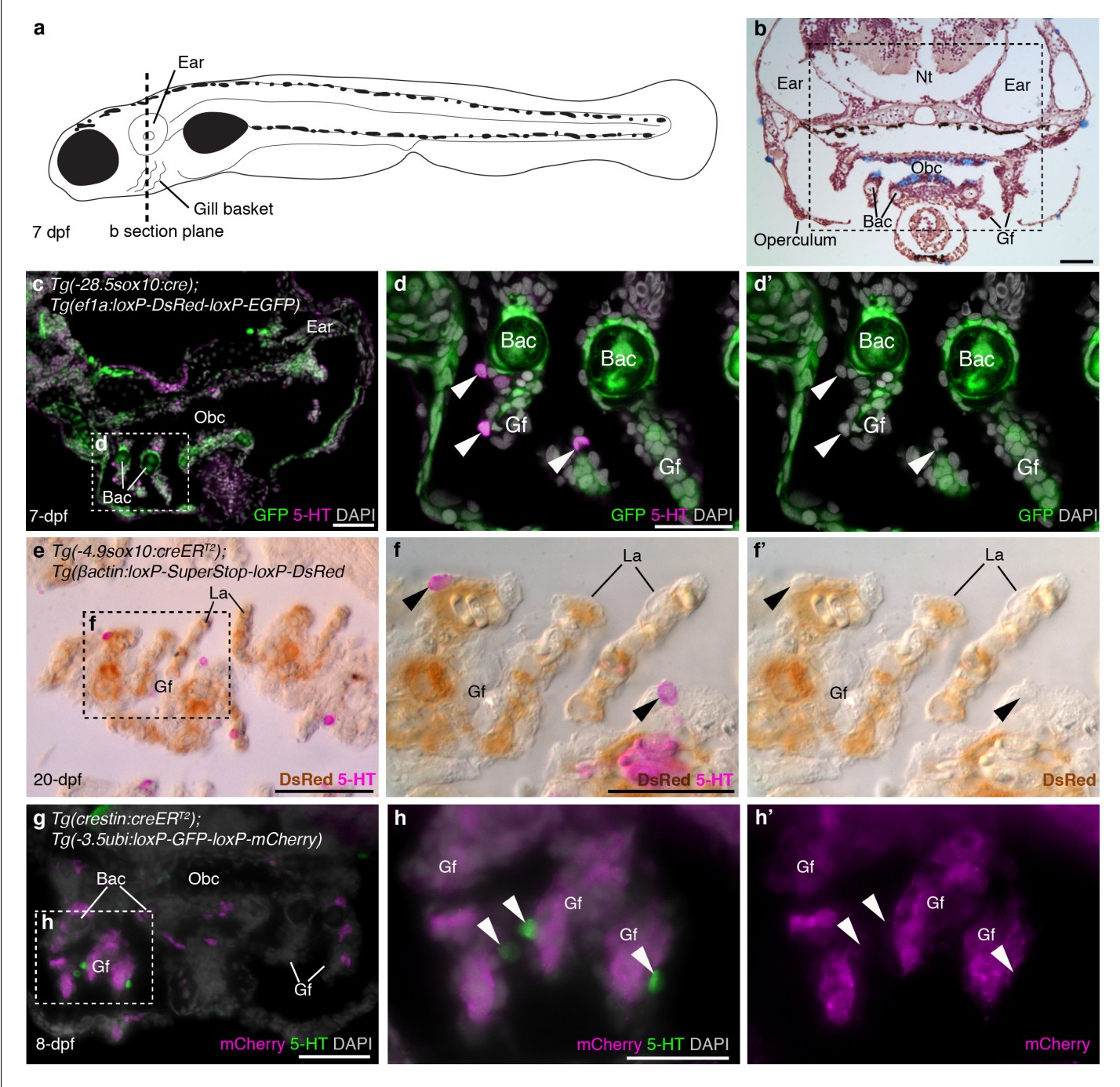

**Figure 1.** Zebrafish NECs are not neural crest-derived: genetic lineage-tracing data. (**a**) Schematic of a 7–8 dpf zebrafish; dotted line indicates section plane in **b-f'**. (**b**) Hematoxylin and eosin staining at 7-dpf reveals gill filaments branching from branchial arch cartilages, and the orobranchial cavity. Dashed box indicates approximate region in **c,g**. (**c–d'**) In 7-dpf *Tg(-28.5sox10:cre);Tg(ef1a:loxP-DsRed-loxP-EGFP)* zebrafish, GFP labels neural crest-derived branchial arch cartilage and mesenchyme, but not NECs in the gill filaments (identified by immunoreactivity for serotonin, 5-HT; arrowheads). (**e–f'**) Horizontal section through the gills of a 20-dpf *Tg(-4.9sox10:creER^{T2});Tg(βactin:loxP-SuperStop-loxP-DsRed)* zebrafish. DsRed (brown precipitate) labels neural crest-derived gill pillar cells, but not NECs (arrowheads; inverted fluorescent image overlaid on bright-field image). (**g–h'**) In 8-dpf *Tg(crestin:creER^{T2});Tg(-3.5ubi:loxP-GFP-loxP-mCherry)* zebrafish, mCherry labels gill pillar cells but not NECs (arrowheads). 5-HT, serotonin; Bac, branchial arch cartilage; Gf, gill filament; La, lamellae; Nt, neural tube; Obc, orobranchial cavity. Scale-bars: 50 μm in **b,c,e,g**; 25 μm in **d,f,h**.

The following figure supplement is available for figure 1:

**Figure supplement 1.** The promoters driving the Cre-switchable reporter cassettes are active in NECs.

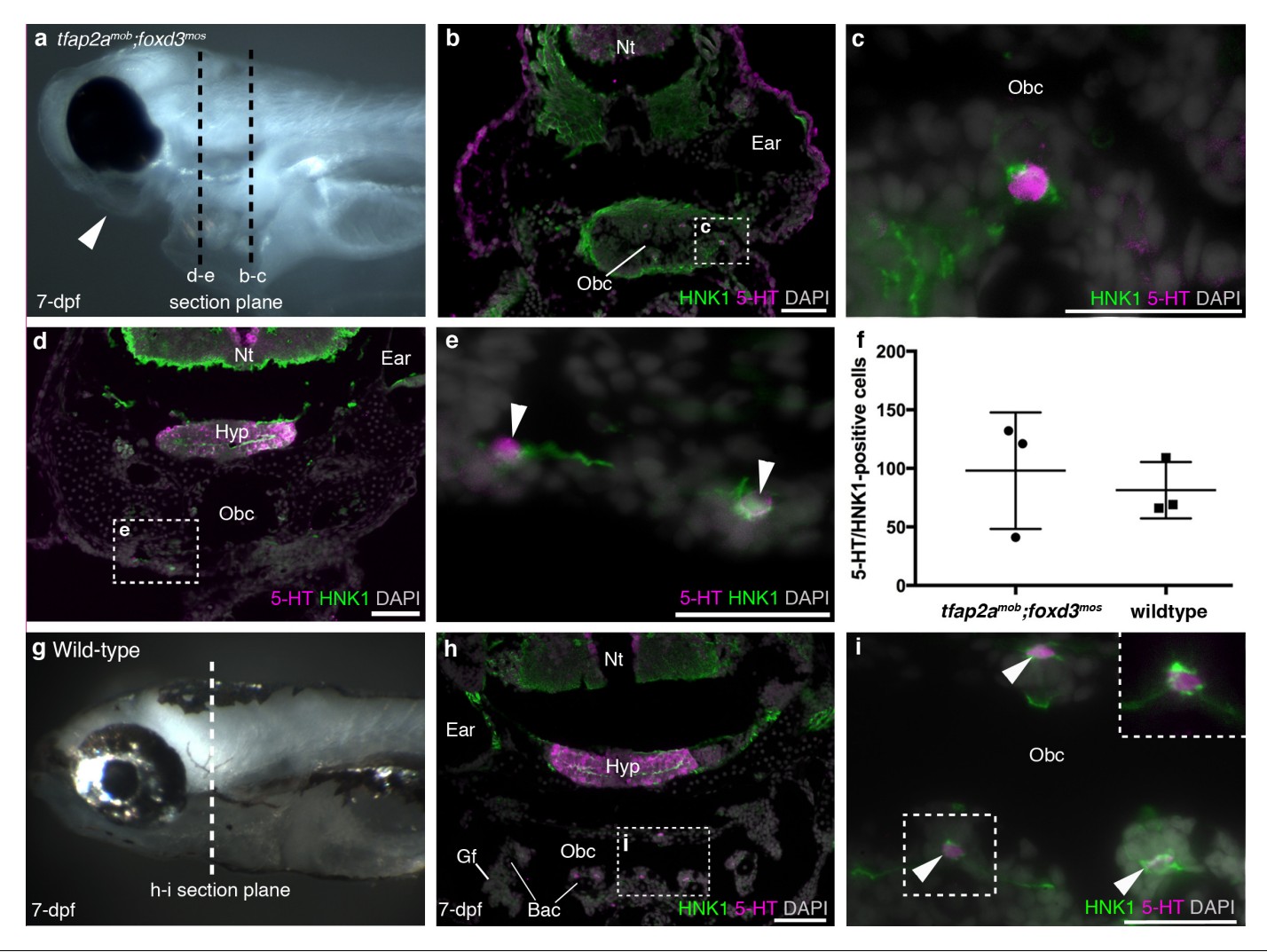

**Figure 2.** Zebrafish NECs are not neural crest-derived: analysis of neural crest-deficient zebrafish mutants. (a) 7-dpf *tfap2a^mob*;*foxd3^mos* zebrafish lack all neural crest derivatives, including melanophores and jaw skeleton (arrowhead). Dotted lines: section planes in **b-e**. (**b,c**) At 7-dpf, *tfap2a^mob*;*foxd3^mos* orobranchial epithelium retains innervated serotonergic (5-HT$^+$) cells (putative NECs), identified by cytoplasmic serotonin surrounded by a ring of HNK1 epitope-immunoreactive neurites. (**d,e**) At 7-dpf, putative NECs (arrowheads) persist in *tfap2a^mob*;*foxd3^mos* zebrafish in the region ventral to the orobranchial cavity where the pharyngeal arches would be located in wild-type fish. (NB The hypothalamus [Hyp] extends caudally beneath the midbrain and rostral hindbrain, and often separates from the overlying brain on sections, as here.) (**f**) The mean number per 7-dpf larva of putative NECs in the orobranchial epithelium does not differ between *tfap2a^mob*;*foxd3^mos* (98.0 ± 49.7 s.d.; n = 3) and wild-type zebrafish (81.3 ± 24.0 s.d.; n = 3) (p=0.63, unpaired two-tailed Student's t-test). All such cells in the orobranchial epithelium were counted for each embryo. Error bars indicate s.d. (**g–i**) Wild-type sibling at 7-dpf. Dotted line: section plane in **h-i**. Putative NECs are present in the orobranchial epithelium (arrowheads; dashed box in **i**, magnified without DAPI in top right corner). 5-HT, serotonin; Gf, gill filament; Hyp, hypothalamus; Nt, neural tube; Obc, orobranchial cavity. Scale-bars: 50 µm in **b,d,h**; 25 µm in **c,e,i**.

The following figure supplement is available for figure 2:

**Figure supplement 1.** Putative NECs in the skin of embryonic zebrafish are not neural crest-derived.

through the skin as well as the gills (*Regan et al., 2011*). Although they have not been shown directly (e.g., by patch-clamp experiments) to be hypoxia-sensitive, the putative NECs in killifish increase in area in response to hypoxia (*Regan et al., 2011*), while in zebrafish, hypoxia decreased or delayed, and hyperoxia accelerated, the normal decline in number of these cells seen with increasing age (*Coccimiglio and Jonz, 2012*). In zebrafish, these serotonergic cells are most

abundant at 3-dpf, and most evident scattered in the skin over the eyes, yolk-sac and tail (*Coccimiglio and Jonz, 2012*). Whole-mount immunostaining of *Tg(crestin:creER^{T2});Tg(-3.5ubi:loxP-GFP-loxP-mCherry)* embryos at 3-dpf for serotonin and mCherry revealed the expected pattern of scattered serotonergic cells in the epidermis, but none was mCherry-positive (i.e., neural crest-derived) (*Figure 2—figure supplement 1* shows a sample three-dimensional rendering for the eye; for the associated z-stack movies, see *Videos 1* and *2*). We quantified this in the eye, where the corneal endothelium is neural crest-derived, as previously reported for chicken (*Noden, 1978*; *Johnston et al., 1979*), mouse (with a minor mesodermal contribution also noted; *Gage et al., 2005*) and *Xenopus* (*Hu et al., 2013*). At 3-dpf, the zebrafish cornea comprises an outer corneal epithelium, a thin acellular collagenous stroma, and a monolayer of flattened corneal endothelial cells (*Soules and Link, 2005*; *Zhao et al., 2006*; *Akhtar et al., 2008*). Across 13 embryos, we counted 328 serotonergic cells in the epidermis over the eye, all of which were mCherry-negative, and 219 nearby mCherry-positive (i.e., neural crest-derived) corneal endothelial cells. Thus, the putative NECs in the skin of embryonic zebrafish are not neural crest-derived.

Taken together, these results show that in zebrafish (a ray-finned fish), the neural crest does not contribute to gill NECs (the previously proposed homologs of glomus cells), or to putative NECs in the orobranchial epithelium and embryonic epidermis.

## Putative NECs in *Xenopus* and lamprey are not neural crest-derived

Innervated serotonergic cells in the internal gills of *Xenopus* tadpoles are the proposed homologs of teleost gill NECs (*Saltys et al., 2006*). We detected scattered serotonergic cells in gill filaments from stage 43 (*Figure 3a,b*), and in the orobranchial epithelium from stage 41 (*Figure 3a,c*). To determine if these putative NECs are neural crest-derived, we unilaterally grafted neural folds from GFP-labeled donors to unlabeled hosts (*Figure 3d*). GFP-labeled neural crest cells migrated away

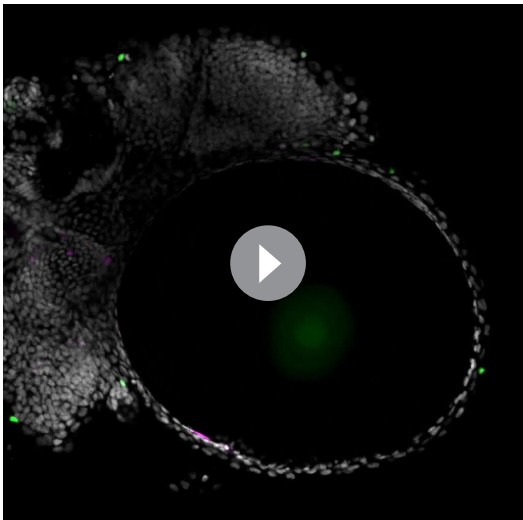

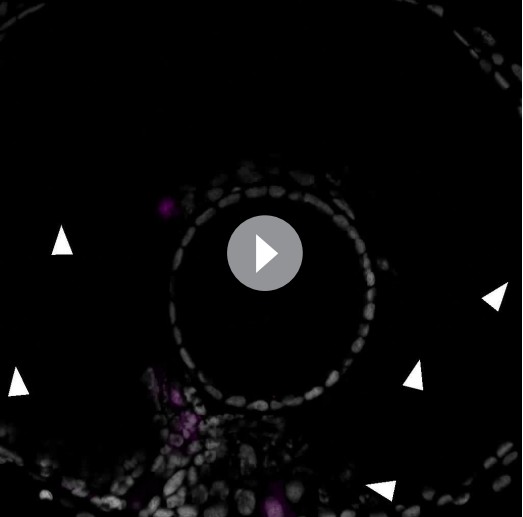

**Video 1.** Putative NECs in the skin of embryonic zebrafish are not neural crest-derived. Z-stack movie showing a whole-mount view of the eye region of a 3-dpf *Tg(crestin:creER^{T2});Tg(-3.5ubi:loxP-GFP-loxP-mCherry)* embryo, immunostained for serotonin and mCherry. Serotonergic cells are scattered in the epidermis over the eye, as reported (*Coccimiglio and Jonz, 2012*), but none is mCherry-positive, i.e., neural crest-derived. *Video 2* shows a higher-power movie. A three-dimensional rendering for the eye is shown in *Figure 2—figure supplement 1*.

**Video 2.** Putative NECs in the skin of embryonic zebrafish are not neural crest-derived. Z-stack movie at higher powerthan *Video 1*, showing a whole-mount view of the eye region of a 3-dpf*Tg*(*crestin:creER^{T2}*);*Tg*(*-3.5ubi:loxP-GFP-loxP-mCherry*)embryo, immunostained for serotonin and mCherry. Serotonergic cells (arrowheads) are scattered in the epidermis over the eye, as reported (*Coccimiglio and Jonz, 2012*), but none is mCherry-positive, i.e., neural crest-derived. A 3-dimensional rendering for the eye is shown in *Figure 2—figure supplement 1*.

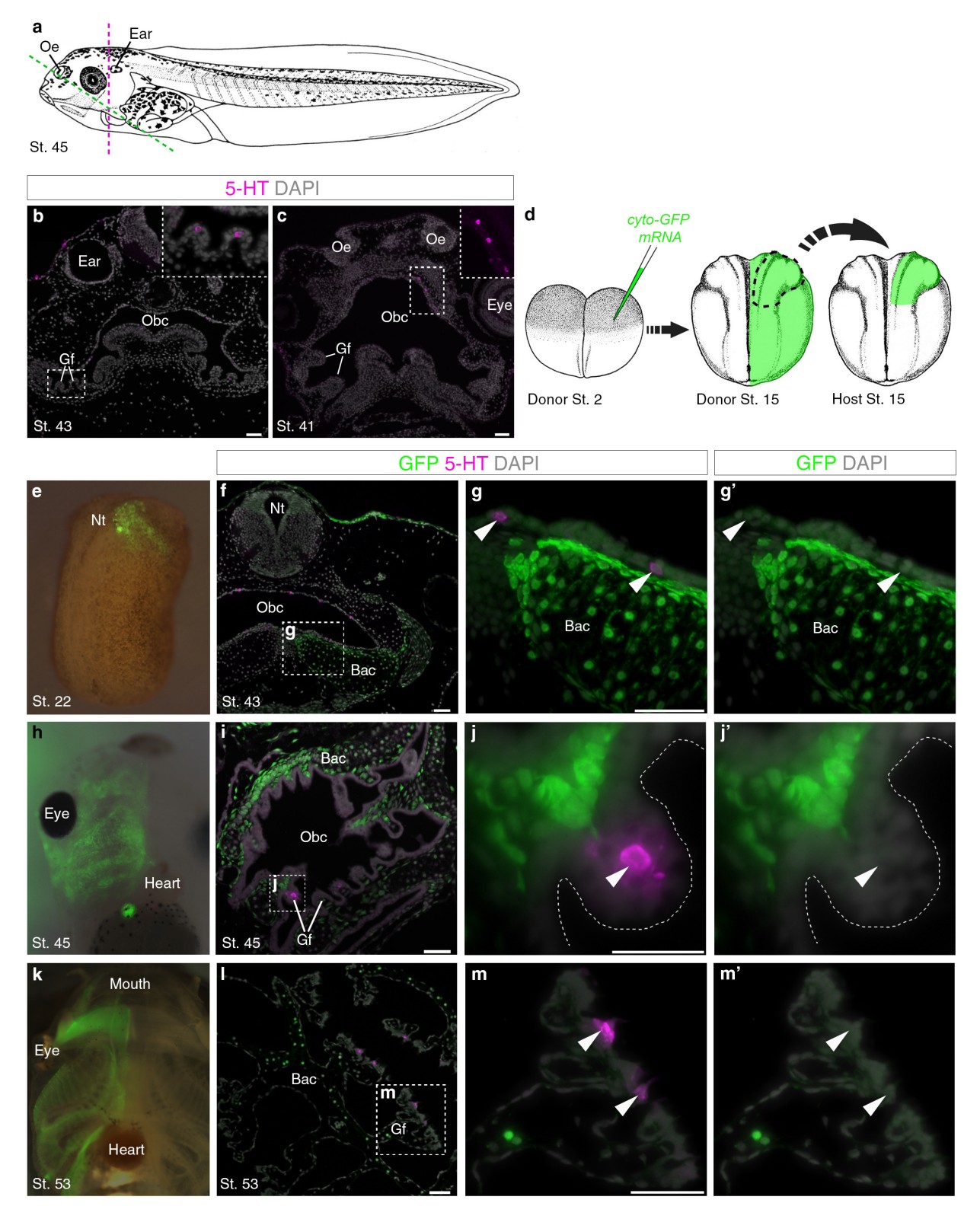

**Figure 3.** Putative *Xenopus* NECs are not neural crest-derived. (a) Schematic of a stage 45 tadpole (***Nieuwkoop and Faber, 1967***). Dotted lines: section planes in **b,f–g'** (magenta, transverse) and in **c,i-j',l-m'** (green, oblique). (b) Serotonergic cells in gill filament epithelium (putative NECs) are first detected at stage 43 (inset shows higher-power view). (c) Similar serotonergic cells are scattered in the orobranchial epithelium from stage 41 (inset shows higher-power view). (d) Schematic (modified from ***Nieuwkoop and Faber, 1967***) showing neural crest labelling: GFP-donors, created by
*Figure 3 continued on next page*

*Figure 3 continued*

injecting *cyto-GFP* mRNA into one cell at the two-cell stage, were grown to stages 13–17, and neural folds grafted unilaterally to wild-type hosts. For grafted embryos grown to stage 53, donors were transgenic *CMV-GFP* embryos. (e) At stage 22, GFP labels neural crest cells migrating towards the branchial arches. (f–g′) At stage 43, GFP labels branchial arch cartilage and surrounding mesenchyme, but not putative NECs (arrowheads) in the orobranchial epithelium. (h–j′) At stage 45, GFP-positive neural crest cells are visible in the branchial arches (whole-mount and section from different embryos). GFP labels branchial arch cartilage and mesenchyme, but not putative NECs (arrowheads) in the gill filaments. (k–m′) At stage 53 (transgenic *CMV-GFP* donors; whole-mount and section from different embryos), GFP labels branchial arch cartilage and mesenchyme, but not putative NECs (arrowheads) in the gill filaments. 5-HT, serotonin; Bac, branchial arch cartilage; Gf, gill filament; Nt, neural tube; Obc, orobranchial cavity; Oe, olfactory epithelium. Scale-bar: 50 μm.

from the neural tube and contributed to branchial arch cartilages and mesenchyme, but putative NECs in the gill filaments and orobranchial epithelium were GFP-negative (n = 14; *Figure 3e–m′*).

In the sea lamprey, the gills are arranged in pairs within the orobranchial cavity, supported by an interbranchial septum (*Figure 4a–c*). Serotonergic cells in lamprey gills are proposed to correspond to the gill NECs of jawed fishes (*Barreiro-Iglesias et al., 2009*). We detected putative NECs from embryonic day (E)18.5, in clusters on the medial edges of the gills, intimately associated with HNK1 epitope-immunoreactive neurites (*Figure 4d*). Scattered serotonergic cells were also found in the epithelium lining the roof and floor of the orobranchial cavity. To determine any neural crest cell contribution, the vital lipophilic dye DiI was injected into E5 vagal neural folds and the embryos followed to E19.0 (*Figure 4e–n*). DiI-labeled neural crest cells migrated into the branchial arches and contributed to the branchial arch basket, as expected (*McCauley and Bronner-Fraser, 2003*), but putative NECs in the gills and orobranchial epithelia, despite being near DiI-labeled cells, were unlabeled (n = 19; *Figure 4i,j,n*; *Figure 4—figure supplement 1*).

These results show that putative NECs in the internal gills and orobranchial epithelium of a frog (i. e., a lobe-finned tetrapod) and the sea lamprey (a jawless fish) are not neural crest-derived.

## Gill NECs are endoderm-derived, like PNECs

Overall, our data show that the neural crest does not contribute to gill NECs in zebrafish, or to their presumed homologues in *Xenopus* and lamprey gills (or to the putative NECs identified in the orobranchial epithelium of all three species). Hence, glomus cells and gill NECs cannot have evolved from the same ancestral cell population. In zebrafish, *Xenopus* and the little skate (a cartilaginous fish), vital dye fate-mapping experiments have shown that the gills and orobranchial cavity are lined with an epithelium derived mostly from endoderm (*Warga and Nüsslein-Volhard, 1999*; *Chalmers and Slack, 2000*; *Gillis and Tidswell, 2017*), suggesting endoderm as an alternative origin for NECs. Indeed, in one of the first descriptions of fish gill NECs (*Dunel-Erb et al., 1982*), they were likened to the PNECs of amniotes, which share a common embryonic origin with other airway epithelial cells in rodents (*Hoyt et al., 1990*; *Rawlins et al., 2009*; *Song et al., 2012*; *Kuo and Krasnow, 2015*). We demonstrated the endodermal origin of PNECs in the mouse by lineage-tracing using the $Sox17^{2A-iCre}$ driver line (in which all endoderm-derived lineages, as well as vascular endothelial cells and the hematopoietic system, express Cre; *Engert et al., 2009*) crossed to the $R26R^{lacZ}$ or $R26R^{tdTomato}$ reporter lines (*Soriano, 1999*; *Madisen et al., 2010*) (*Figure 5a–e′′′*). We also confirmed the recent exclusion by *Wnt1-Cre* lineage-tracing (*Danielian et al., 1998*) of a neural crest contribution to mouse PNECs (*Kuo and Krasnow, 2015*) (*Figure 5—figure supplement 1a–b′*). Similarly, we saw no neural crest contribution to PNECs in the chicken lung after labeling the premigratory neural crest by neural fold grafting from GFP-transgenic donor embryos (*McGrew et al., 2008*) (*Figure 5—figure supplement 1c–e′*).

To test the hypothesis that NECs, like PNECs, are endoderm-derived, we first attempted to label the pharyngeal endoderm of *Xenopus* embryos at stage 14 via focal DiI injections into anterior endoderm (*Chalmers and Slack, 2000*) (*Figure 5f–i*). Only three embryos with endoderm-specific DiI labeling survived to stage 45 for analysis. The DiI labeling was very sparse, but in one embryo, 15 serotonergic cells in the orobranchial epithelium (putative NECs) were DiI-labeled, supporting an endodermal origin (*Figure 5g–i′′*).

Since the direct labeling approach in *Xenopus* proved to be technically challenging, we used genetic lineage-tracing of endodermal *sox17* expression in zebrafish, inducing gastrulation-stage

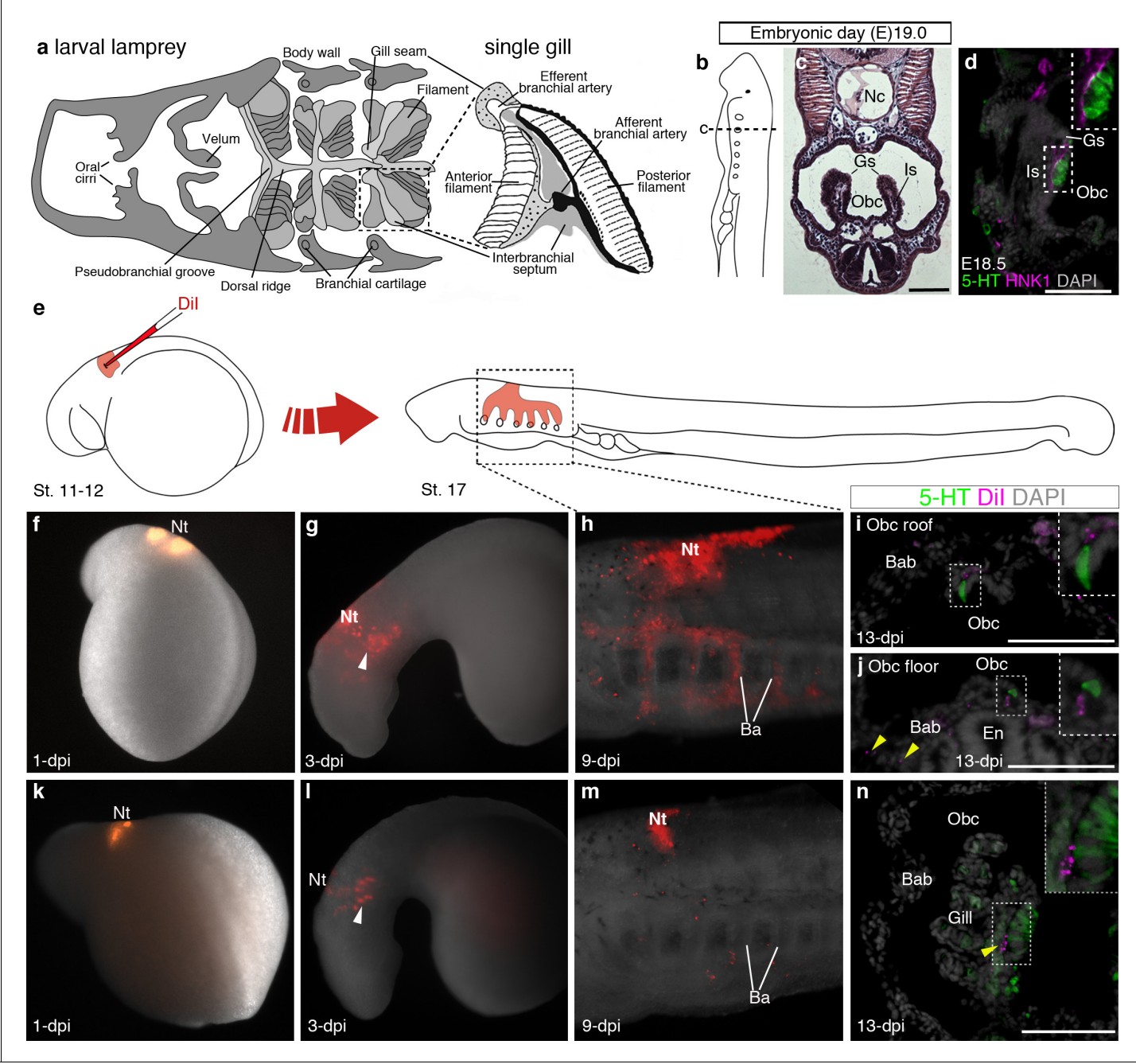

**Figure 4.** Putative lamprey NECs are not neural crest-derived. (**a**) Schematic larval lamprey section (modified from *Barreiro-Iglesias et al., 2009*) showing gill pairs in the orobranchial cavity, supported by an interbranchial septum, and a single gill at higher power. (**b**) Schematic Piavis-stage 17 (E19) lamprey (modified from *Tahara, 1988*). Dotted line shows section plane in **c,d,i,j,n**. (**c**) Hematoxylin and eosin staining at E19 shows internal gills as 'stalks' within the orobranchial cavity. (**d**) Putative NECs (serotonergic cells associated with HNK1 epitope-immunoreactive neurites) are first visible at E18.5 in the medial gill epithelium. (**e**) Schematic (modified from *Tahara, 1988*) showing neural crest labelling by DiI injection at E5 (Piavis stages 11–12). (**f–n**) Two different embryos (**f–j, k–n**), showing DiI-labeled neural crest cells migrating ventrally (arrowheads, **g,l**) into the branchial arches, contributing to branchial arch basket and gill supporting cells (arrowheads, **j,n**), but not serotonergic cells (putative NECs) in the orobranchial epithelium (**i,j**) or gills (**n**). 5-HT, serotonin; Ba, branchial arch; Bab, branchial arch basket; En, endostyle; Gs, gill seam; Is, interbranchial septum; Nc, notochord; Nt, neural tube; Obc, orobranchial cavity. Scale-bar: 50 μm.

The following figure supplement is available for figure 4:

**Figure supplement 1.** Confirmation of successful targeting of pharyngeal arch-destined neural crest cells in all lamprey embryos analyzed for neural crest contribution to putative NECs.

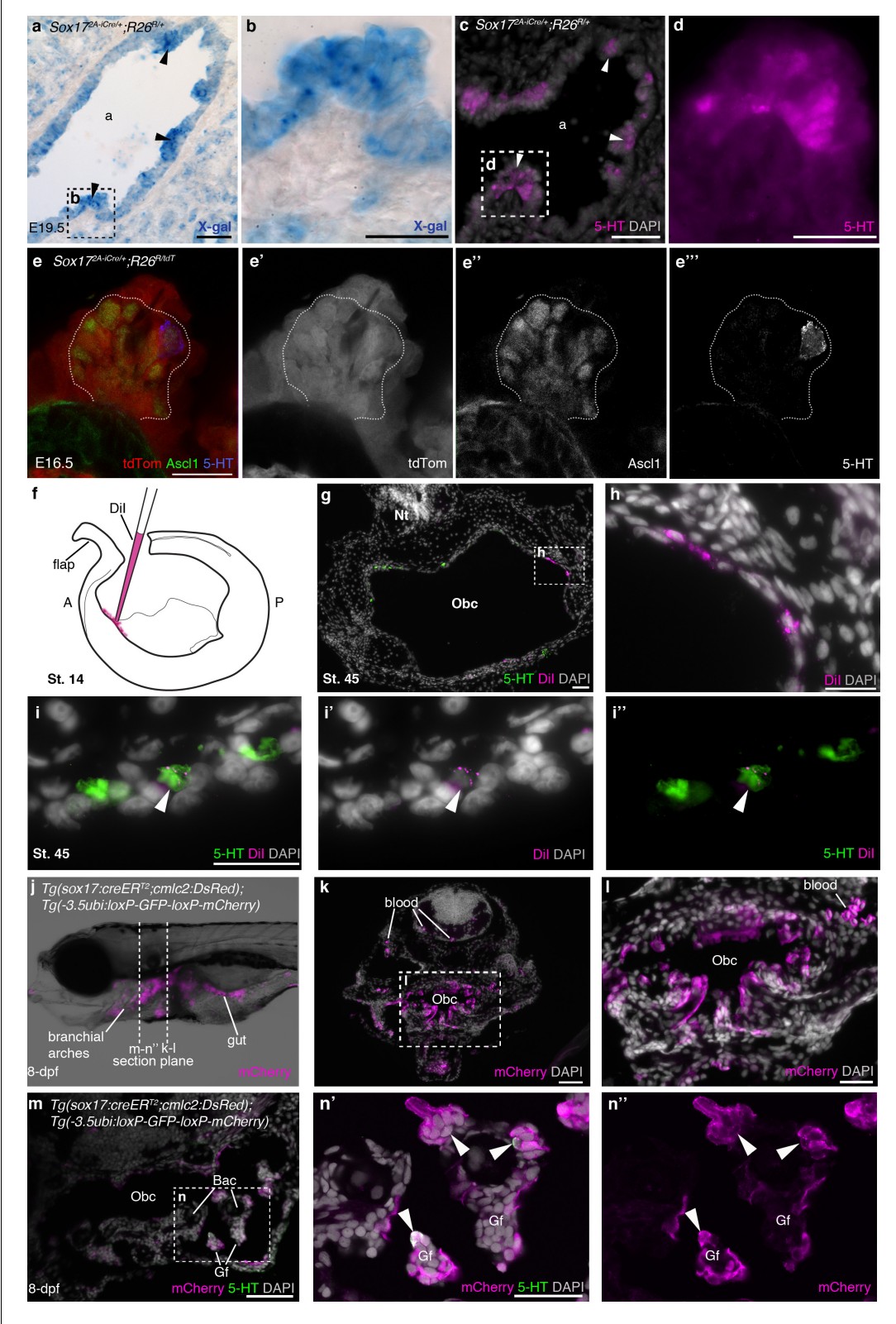

**Figure 5.** NECs are endoderm-derived, like PNECs. (**a–d**) In *Sox17²ᴬ⁻ⁱᶜʳᵉ/⁺;R26ᴿ/⁺* mice, all endoderm-derived lineages, as well as vascular endothelial cells and the hematopoietic system, constitutively express *β*-galactosidase (***Engert et al., 2009***). Serial sections of an E19.5 *Sox17²ᴬ⁻ⁱᶜʳᵉ/⁺;R26ᴿ/⁺* mouse lung show that X-gal labels PNECs (**a,b**; black arrowheads), whose identity is confirmed by serotonin expression (**c,d**; white arrowheads). The serotonin-positive cells are clearly all in the epithelium, which is entirely X-gal-positive, although there is some variation in staining level from cell to cell. (**e–e'''**) A

*Figure 5 continued on next page*

*Figure 5 continued*

high-power view of a cluster of Ascl1-expressing PNECs in a section of an E16.5 *Sox17²ᴬ⁻ⁱᶜʳᵉ/⁺;R26Rᵗᵈᵀᵒᵐᵃᵗᵒ* mouse lung, in which endoderm-derived lineages express tdTomato. Only the occasional PNEC is serotonin-positive at this stage. The Ascl1-expressing PNECs are tdTomato-positive, i.e., endoderm-derived. (**f–i''**) An endodermal contribution to putative NECs in *Xenopus* was investigated by performing focal DiI injections into the anterior endoderm at stage 14 (**f**), as described in *Chalmers and Slack (2000)*. At stage 45, DiI labels the endoderm lining the orobranchial cavity (**g,h**), and serotonergic cells (putative NECs, arrowheads) in the orobranchial epithelium (**i–i''**). (**j–l**) In *Tg(sox17:creERᵀ²;cmlc2:DsRed);Tg(-3.5ubi:loxP-GFP-loxP-mCherry)* zebrafish, the endoderm is labeled with mCherry and (**m–n''**) NECs in the gill filaments are mCherry-positive (arrowheads). 5-HT, serotonin; A, anterior; a, airway; Bac, branchial arch cartilage; Gf, gill filament; Obc, orobranchial cavity; P, posterior; tdTom, tdTomato. Scale-bars: 50 μm in **a,c,g, k,m**; 25 μm in **b,d,h,i,l,n**; 20 μm in **e**.

The following figure supplement is available for figure 5:

**Figure supplement 1.** The neural crest does not contribute to amniote PNECs.

Cre expression and recombination in embryos from crosses between a *sox17:creERᵀ²* zebrafish driver line [*Tg(sox17:creERᵀ²;cmlc2:DsRed)*; Joseph J. Lancman, Keith P. Gates, and P. Duc S. Dong, personal communication, March, 2017] and the switchable reporter line *Tg(-3.5ubi:loxP-GFP-loxP-mCherry)* (*Mosimann et al., 2011*) (*Figure 5j–n''*). At 8-dpf, mCherry expression was seen in both gill NECs and putative NECs in the orobranchial epithelium: 147/331 serotonergic cells counted across six larvae (≥36 cells counted per fish) were mCherry-positive (*Figure 5n–n''*). [The relatively low labeling efficiency likely results from a lack of optimization of the 4-OHT dose for the *Tg(sox17: creERᵀ²;cmlc2:DsRed)* driver in combination with this particular switchable reporter line (*Mosimann et al., 2011*).] These data demonstrate an endodermal origin in zebrafish for gill NECs, and also for putative NECs in the orobranchial epithelium. (Our genetic lineage-tracing data also confirm the endodermal origin of zebrafish gill filament epithelium, previously reported from vital dye fate-mapping experiments; *Warga and Nüsslein-Volhard, 1999*.)

Taken together, these results reveal the endodermal origin of gill NECs and putative NECs in the orobranchial epithelium. This supports the shared evolutionary ancestry of NECs with endoderm-derived PNECs, rather than neural crest-derived glomus cells.

## NECs do not express Phox2b, which is essential for glomus cell development

It is formally possible that, despite their different embryonic origins, glomus cells and NECs could still be homologous cell types through activation of the same genetic network. Although the molecular basis of NEC development has not been investigated, the basic helix-loop-helix transcription factor Ascl1 (Mash1) is required for the formation of both PNECs (*Ito et al., 2000*) and glomus cells (*Kameda, 2005*). The homeodomain transcription factor Phox2b is also essential for glomus cell development (*Dauger et al., 2003*). However, we were unable to detect Phox2b-positive cells in embryonic zebrafish gills or orobranchial epithelium at 5-dpf (n = 4) or 7-dpf (n = 3), although Phox2b was expressed by a subset of cells in the hindbrain, as expected (*Coppola et al., 2012*) (*Figure 6a–f'*). Similarly, we could not detect any *Phox2*-positive cells in sea lamprey gills or orobranchial epithelia at E16 or E18 (n = 6), although *Phox2* was expressed in the epibranchial ganglia, and in patches of ectoderm and subjacent mesenchyme ventral to the epibranchial ganglia (*Figure 6g–j'*), in the same position as the hypobranchial placodes and associated ganglia identified in *Xenopus* (*Schlosser, 2003*).

We also found that PNECs lack Phox2b expression. In mouse embryos at E16.5, when PNECs can be identified by Ascl1 expression (serotonin is only expressed in a few PNECs at this stage), Phox2b was not seen in PNECs, despite expression in nearby intrinsic pulmonary ganglia (n = 2; *Figure 6k–l'*). Similarly, in chicken embryos at E12-E13.5, when scattered PNECs can be identified in the lung epithelium by serotonin immunoreactivity, Phox2b expression was not seen in the lung epithelium either by in situ hybridization or by immunostaining, although it could be detected in nearby intrinsic pulmonary ganglia (n = 3; *Figure 6m,m'*).

Overall, these data show that NECs (and PNECs) do not activate the same genetic network as glomus cells, since they lack expression of a transcription factor, Phox2b, which is essential for

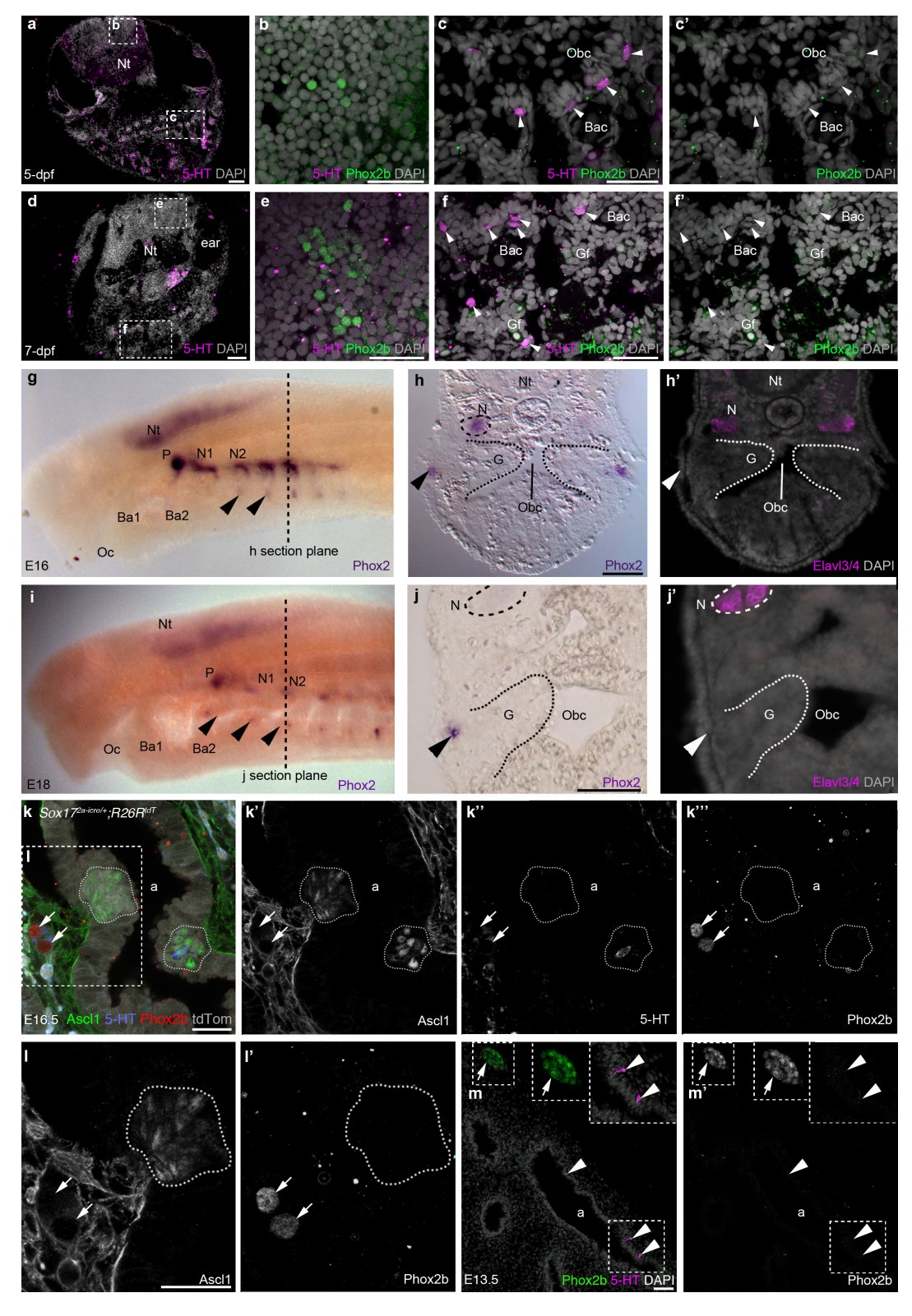

**Figure 6.** Phox2b expression is absent from gill and lung epithelia. (a–f') In wild-type zebrafish at 5- and 7-dpf, Phox2b is expressed by a subset of cells in the hindbrain (b,e), but not by gill NECs or putative NECs in the orobranchial epithelium (arrowheads; c–c',f–f'). (g–j') At E16 (g–h') and E18 (i–j') in the sea lamprey, *Phox2* expression is seen in the neural tube, the epibranchial (petrosal and nodose) ganglia (identified in section by the neuronal marker Elavl3/4), and in patches of ectoderm and subjacent mesenchyme ventral to the epibranchial ganglia (arrowheads). However, *Phox2* expression

*Figure 6 continued on next page*

*Figure 6 continued*

is absent from the gill epithelium, where putative NECS would be located. Dotted lines in panels **g** and **i** indicate the section plane in **h** and **j**. (**k–l'**) In a section of an E16.5 *Sox17*<sup>2A-iCre/+</sup>;*R26*<sup>tdTomato</sup> mouse lung, Phox2b expression is seen in intrinsic pulmonary ganglia (arrows), but not in Ascl1/serotonin-positive PNECs located in the lung airway epithelium (dotted lines outline clusters of PNECs). (**m,m'**) In a section of an E13.5 chicken lung, Phox2b expression is seen in an intrinsic pulmonary ganglion (arrow), but not in serotonin-positive PNECs scattered in the lung airway epithelium (arrowheads). Insets show higher power views. a, airway; Ba, branchial arch; Bac, branchial arch cartilage; G, gill; Gf, gill filament; N, nodose ganglion; Nt, neural tube; Obc, orobranchial cavity; Oc, oral cavity; P, petrosal ganglion. Scale bars: 50 µm in **a,d,h,j,m**; 25 µm in **b,c,e,f,k,l**.

glomus cell development (*Dauger et al., 2003*). Hence, NECs and glomus cells cannot be homologous cell types.

## Candidate glomus cell homologues in fish: neural crest-derived chromaffin cells associated with pharyngeal arch blood vessels

Since NECs are endoderm-derived, we reasoned that neural crest-derived glomus cells must have evolved independently. Glomus cells are catecholaminergic, like the neural crest-derived chromaffin cells of the adrenal gland, which are also hypoxia-sensitive (reviewed by *López-Barneo et al., 2016*). Intriguingly, in lampreys, catecholaminergic (chromium salt-staining, i.e., 'chromaffin') cells were reported a century ago in association with large blood vessels not only in the trunk, but also as far rostrally as the second branchial arch, 'in the walls of the segmental veins as these run round the notochord' (*Giacomini, 1902*; *Gaskell, 1912*). This suggested to us the possibility that glomus cells could have evolved from catecholaminergic cells associated with blood vessels in anamniote pharyngeal arches, if such cells are neural crest-derived.

We first used immunostaining for the catecholaminergic marker tyrosine hydroxylase and the neurite-marker acetylated tubulin to confirm the existence of catecholaminergic cells, at least some of which may be innervated, in the walls of the anterior cardinal veins in the gill arches of ammocoete-stage sea lamprey (*Figure 7a–c*). It would be difficult to test whether the gill arch catecholaminergic cells are neural crest-derived: even if DiI-labeled embryos could be raised to ammocoete stages, the DiI-labeling would likely be very sparse. We reasoned that if present in lamprey, such cells might also be present in the zebrafish. We sectioned metamorphic juveniles and identified similar clusters of catecholaminergic cells in close association with blood vessels in the gill arches (*Figure 7d–e'''*). To our knowledge, this is the first demonstration of the existence of catecholaminergic cells associated with gill arch blood vessels in a jawed anamniote. Importantly, these catecholaminergic cells were mCherry-positive, i.e., neural crest-derived, in *Tg(crestin:creER*<sup>T2</sup>*);Tg(-3.5ubi:loxP-GFP-loxP-mCherry)* metamorphic juveniles (*Figure 7e–e'''*) (51 such catecholaminergic cells were mCherry-positive across five juveniles [≥5 counted per fish]). This discovery suggests a new, speculative hypothesis for carotid body evolution, namely that it evolved in the amniote lineage via the aggregation of neural crest-derived catecholaminergic cells that were already associated with pharyngeal arch blood vessels in anamniotes.

## Discussion

### Gill NECs are endoderm-derived, suggesting shared ancestry with PNECs, not glomus cells

The evolutionary history of the hypoxia-sensitive cells that initiate amniote respiratory reflexes has been obscure, but is critical for our understanding of the transition from aquatic to terrestrial life. This involved a change from aquatic respiration in an environment with low oxygen solubility, to obligate air-breathing in an environment with more stable oxygen levels. It has been proposed that this transition was accompanied by a switch from a dispersed population of externally oriented hypoxia-sensitive cells in the gills that monitored the highly variable external oxygen levels, to one dominant site of hypoxia-sensitive cells that focused on monitoring internal oxygen states (*Burleson and Milsom, 2003*; *Milsom and Burleson, 2007*). Current hypotheses suggest that this change was accompanied by the evolution of the glomus cells of the carotid body from an ancestral population of gill NECs (e.g., *Milsom and Burleson, 2007*; *Hempleman and Warburton, 2013*; *Jonz et al., 2016*).

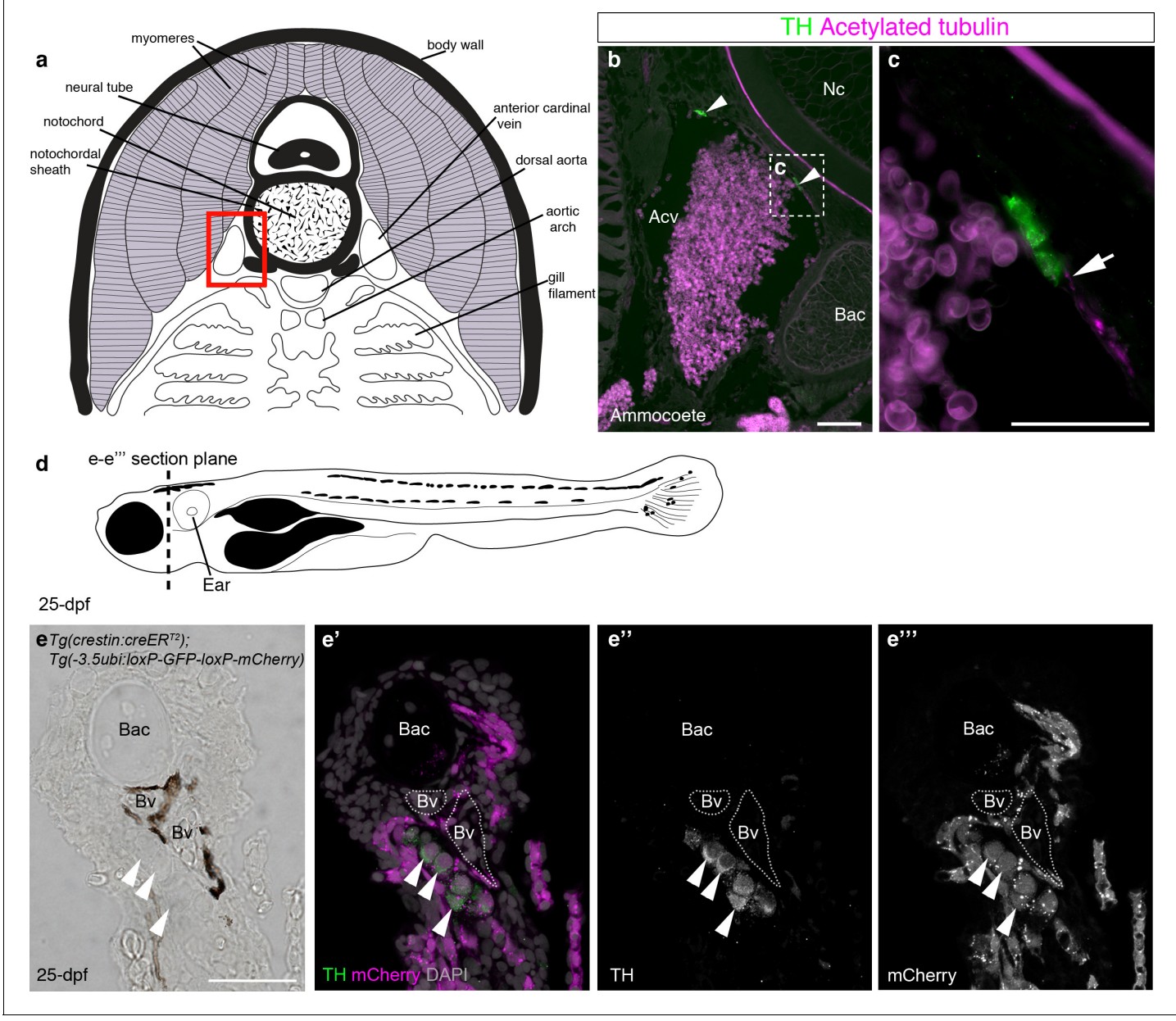

**Figure 7.** Catecholaminergic cells associated with gill arch blood vessels are neural crest-derived in zebrafish. (a) Schematic transverse section through ammocoete-stage lamprey gill arch (modified from *Ruppert et al., 2003*). Red box indicates region shown in **b**. (b,c) Tyrosine hydroxylase-positive (catecholaminergic) cells are present in the wall of the anterior cardinal vein (arrowheads), closely associated with acetylated tubulin-immunoreactive neurites (arrow). (d) Schematic 25-dpf zebrafish. Dotted line indicates transverse section plane through the gill basket in **e-e'''**. (e–e''') Tyrosine hydroxylase-positive (catecholaminergic) cells (arrowheads) seen adjacent to melanocyte-covered gill-filament blood vessels (dotted lines), are neural crest-derived (mCherry-positive) in 25-dpf *Tg(crestin:creER^T2);Tg(-3.5ubi:loxP-GFP-loxP-mCherry)* zebrafish. Acv, anterior cardinal vein; Bac, branchial arch cartilage; Bv, blood vessel; Nc, notochord; TH; tyrosine hydroxylase. Scale-bars: 50 µm in **b**; 25 µm in **c,e**.

This was entirely plausible, given their common association with pharyngeal arch arteries, afferent innervation by glossopharyngeal and/or vagal nerves, and hypoxia-sensitive $K^+$ currents (*López-Barneo et al., 1988*; *Buckler, 1997*; *Jonz et al., 2004*; *Qin et al., 2010*). However, glomus cells are neural crest-derived (*Le Douarin et al., 1972*; *Pearse et al., 1973*; *Pardal et al., 2007*) and the hypothesis is not compatible with our demonstration that neural crest cells do not contribute to gill NECs in zebrafish, or their presumed homologues in *Xenopus* and lamprey, or to similar innervated

serotonergic cells (putative NECs) in the orobranchial epithelium of all three anamniote species. In contrast, we found that these cells are endoderm-derived, like PNECs, which differentiate in situ within pulmonary airway epithelia (*Hoyt et al., 1990*; *Rawlins et al., 2009*; *Song et al., 2012*; *Kuo and Krasnow, 2015*) and whose endodermal origin we demonstrated using *Sox17-Cre* lineage-tracing in mouse. This suggests that gill NECs, and putative NECs in the orobranchial epithelium if these prove to be hypoxia-sensitive, more likely share a common ancestor with PNECs. Denervation experiments in various fishes (a shark, as well as some teleosts) have shown that hypoxia-sensitive and/or $CO_2$-sensitive chemoreceptors involved in ventilation responses are located in the orobranchial cavity, as well as in the gills, although their cellular identity has not been confirmed (reviewed by *Milsom, 2012*). Furthermore, putative NECs (identified by morphology, innervation and serotonin immunoreactivity, although not as yet shown to be hypoxia-sensitive) have been reported in the epithelium of the air-breathing organs (where present) of ray-finned fishes, lobe-finned lungfishes and amphibians (reviewed by *Jonz et al., 2016*; *Hsia et al., 2013*). Anamniote air-breathing organs likely evolved from out-pocketings of the caudal orobranchial epithelium after the evolution of gills and NECs (reviewed by *Hsia et al., 2013*). Taken together, this leads us to speculate that hypoxia-sensitive NECs in the epithelia of the gills and orobranchial cavity of ancestral vertebrates were retained in the air-breathing organs of both anamniotes and amniotes. Testing this hypothesis for the evolutionary origin of PNECs must await evidence for the hypoxia-sensitivity of the putative NECs in the orobranchial epithelium of anamniotes.

Putative NECs have also been identified via serotonin immunoreactivity in the skin of developing zebrafish (*Jonz and Nurse, 2006*; *Coccimiglio and Jonz, 2012*) and of adult mangrove killifish, which respire through the skin as well as the gills (*Regan et al., 2011*). Our genetic lineage-tracing data in zebrafish show that these cells are not neural crest-derived. If these serotonergic cells in the embryonic skin indeed prove to be hypoxia-sensitive NECs, then we suggest that NECs are likely a 'local epithelial' rather than uniquely endodermal cell type, i.e., that they can differentiate within epithelia of either ectodermal or endodermal origin, like taste buds (*Barlow and Northcutt, 1995*; *Stone et al., 1995*) and the ameloblast (enamel-forming) layer of teeth (*Soukup et al., 2008*).

It remained formally possible that, despite their different embryonic origins, gill NECs and glomus cells could be homologous cell types via activation of the same genetic network. However, we found that the transcription factor Phox2b, which is critical for glomus cell development (*Dauger et al., 2003*), is not expressed by embryonic zebrafish gill NECs, putative NECs in the orobranchial epithelium, or their putative homologues in lamprey embryos. (We also found that mouse and chicken PNECs lack Phox2b expression.) Hence, NECs and glomus cells cannot be homologous cell types.

## A new hypothesis for carotid body evolution

Since neural crest-derived glomus cells could not have evolved from endoderm-derived gill NECs, what is their evolutionary history? Glomus cells are catecholaminergic, and they are strikingly similar to the neural crest-derived chromaffin (i.e., catecholaminergic) cells of the adrenal medulla. For example, in fetal or neonatal mammals, adrenal chromaffin cells release catecholamines in direct response to hypoxia, like glomus cells (*Comline and Silver, 1966*; *Cheung, 1989*, *1990*; *Seidler and Slotkin, 1985*; *Thompson et al., 1997*). This 'non-neurogenic' response to hypoxia, in the absence of neural input, facilitates the transition to air-breathing in neonates by stimulating lung fluid absorption and regulating cardiovascular function (*Seidler and Slotkin, 1985*; *Thompson et al., 1997*). The sensitivity of adrenal chromaffin cells to hypoxia is lost upon postnatal cholinergic innervation of the adrenal gland, although at least some hypoxia-responsive chromaffin cells persist in the adult adrenal medulla (*García-Fernández et al., 2007*; *Levitsky and López-Barneo, 2009*) (also see *López-Barneo et al., 2016*). Furthermore, hypoxia inhibits $K^+$ currents in adrenal chromaffin cells, as in glomus cells (reviewed by *López-Barneo et al., 2016*), and the 'set point' of hypoxia sensitivity is controlled in both glomus cells and adrenal chromaffin cells by mutual antagonism between the oxygen-regulated transcription factors hypoxia-inducible factor 1-alpha (Hif1α/HIF1a) and hypoxia-inducible factor 2-alpha (Hif2α/HIF2a) (*Yuan et al., 2013*).

These similarities led us to re-visit century-old reports (*Giacomini, 1902*; *Gaskell, 1912*) of chromaffin (chromium salt-staining, i.e., catecholaminergic) cells associated with large branchial arch blood vessels in lamprey. We confirmed the existence of these catecholaminergic cells in ammocoete-stage sea lamprey, and went on to discover catecholaminergic cells associated with pharyngeal arch blood vessels in juvenile zebrafish, whose neural crest origin we demonstrated by genetic

lineage-tracing. We speculate that the carotid body may have evolved via the aggregation of such cells, and their subsequent acquisition of serotonergic properties and afferent innervation by glossopharyngeal and/or vagal afferents, such that they became incorporated into the afferent arm of respiratory reflexes. In order to test this hypothesis, it will be necessary to investigate whether these blood vessel-associated catecholaminergic cells in anamniotes secrete catecholamines in response to hypoxia. Glomus cells in the carotid body are enveloped by glial-like sustentacular cells, also neural crest-derived (*Le Douarin et al., 1972*; *Pearse et al., 1973*; *Pardal et al., 2007*), which have been shown to act as adult stem cells for the production of new glomus cells under hypoxic conditions (*Pardal et al., 2007*). Hypotheses for carotid body evolution must also take these cells into account.

The amphibian carotid labyrinth, a maze-like vascular expansion at the bifurcation of the carotid artery, is sensitive to oxygen levels (*Ishii et al., 1966*) and considered a carotid body homologue (reviewed by *Kusakabe, 2009*). It contains both serotonergic and catecholaminergic cells, innervated by glossopharyngeal and/or vagal nerves (*Reyes et al., 2014*), with both efferent and afferent synapses (reviewed by *Kusakabe, 2009*). Furthermore, the glomus cells in the carotid labyrinth are enveloped by fine processes of sustentacular cells (reviewed by *Kusakabe, 2009*). In adult amphibians, the carotid labyrinth is hypothesized to replace the NECs of the larval gills as the primary site of the chemosensors responsible for maintaining respiratory homeostasis (*Kusakabe, 2002*; *Jonz and Nurse, 2006*). Direct electrophysiological evidence is lacking for which cells are hypoxia-responsive, however, and their embryonic origin has not been established. Further investigation of the development and physiology of the carotid labyrinth in amphibians, and of the neural crest-derived catecholaminergic cells that we discovered in association with pharyngeal arch blood vessels in zebrafish, should help to test our new hypothesis for carotid body evolution.

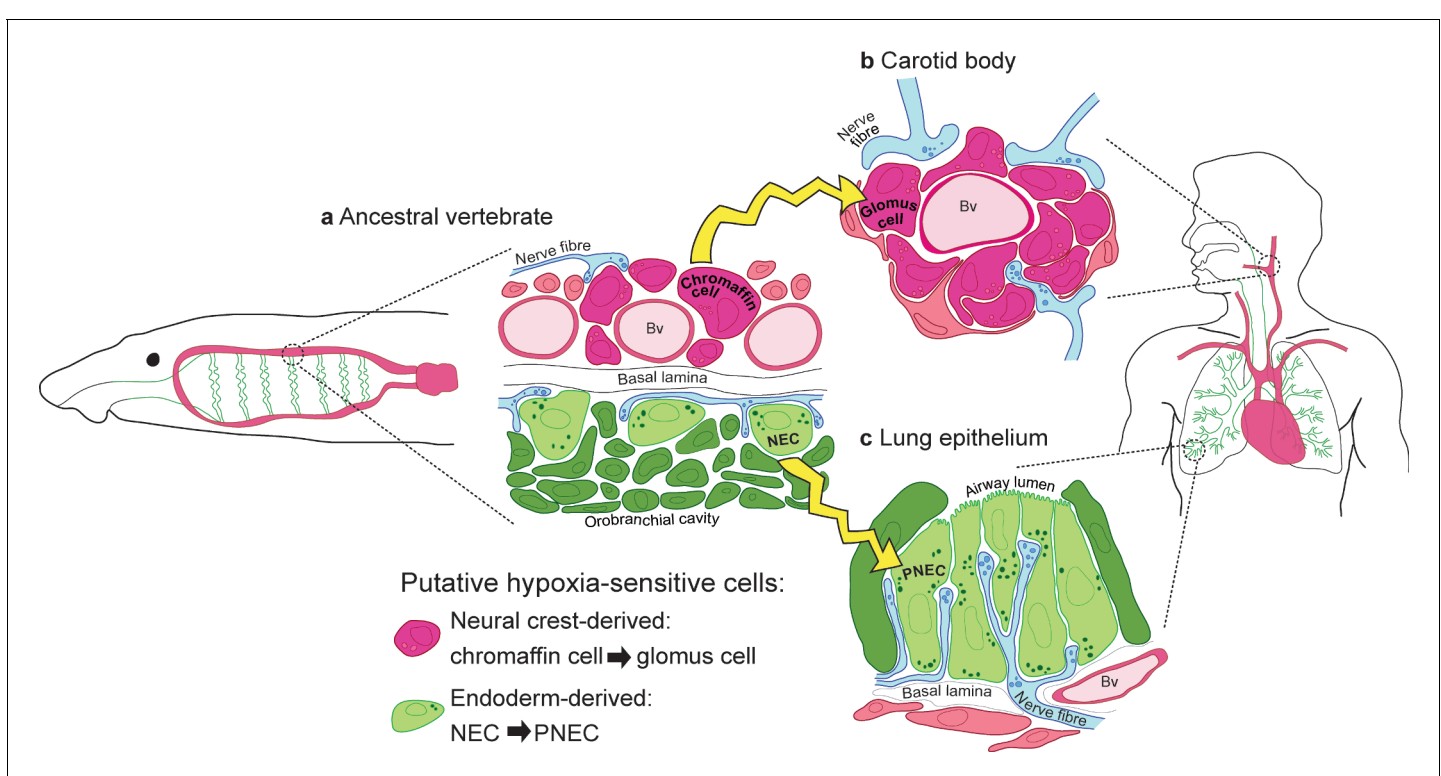

**Figure 8.** Model for the evolution of the hypoxia-sensitive cells involved in amniote respiratory reflexes. (a) Schematic ancestral vertebrate with internal gills. Neural crest-derived (magenta) chromaffin cells are associated with large blood vessels in the pharyngeal arches, while NECs differentiate within the endoderm-derived (green) epithelium lining the gills and orobranchial cavity. (b) During the transition to terrestrial life, the glomus cells of the carotid body evolved via the aggregation of neural crest-derived chromaffin cells (which must also have acquired serotonergic properties), while (c) NECs persisted as PNECs in lung airway epithelia. Yellow arrows indicate shared ancestry. Bv, blood vessel; NEC, neuroepithelial cell; PNEC, pulmonary neuroendocrine cell.

## A new model for amniote hypoxia-sensitive cell evolution

Given our lineage-tracing data, we present a new model for the evolution of the hypoxia-sensitive cells involved in amniote respiratory reflexes (*Figure 8*). We hypothesize that carotid body glomus cells evolved via the aggregation of neural crest-derived catecholaminergic (chromaffin) cells that were already associated with blood vessels in anamniote gill arches (and which must subsequently have acquired serotonergic properties and afferent innervation by glossopharyngeal and/or vagal nerves), while NECs differentiating in situ in the endoderm-derived epithelia of the gills and orobranchial cavity were retained as PNECs in lung airway epithelia. This model can be viewed as more parsimonious, since both embryonic lineage (neural crest versus endoderm) and function (physiological versus environmental oxygen monitoring) are maintained during the proposed evolutionary history of glomus cells and PNECs. Testing the model will require investigation of the physiology and hypoxia-responsiveness of anamniote gill arch blood vessel-associated catecholaminergic cells.

# Materials and methods

## Zebrafish lines

The following zebrafish (*Danio rerio*) lines were used: *Tg(-28.5sox10:cre);Tg(ef1a:loxP-DsRed-loxP-EGFP)* (*Kague et al., 2012*), *Tg(-4.9sox10:creER^{T2});Tg(βactin:loxP-SuperStop-loxP-DsRed)* (*Mongera et al., 2013*), *Tg(crestin:creER^{T2});Tg(-3.5ubi:loxP-GFP-loxP-mCherry)* (*Mosimann et al., 2011*; *Kaufman et al., 2016*), *tfap2a^{mob};foxd3^{mos}* (*Wang et al., 2011*) and *Tg(sox17:creER^{T2};cmlc2:DsRed)* [created using the *sox17* promoter from *Mizoguchi et al. (2008)*; Joseph J. Lancman, Keith P. Gates, and P. Duc S. Dong, personal communication, March, 2017]. Experiments using *Tg(-4.9sox10:creER^{T2});Tg(βactin:loxP-SuperStop-loxP-DsRed)* zebrafish were conducted in compliance with the regulations of the Regierungspräsidium Tübingen and the Max Planck Society. Experiments using all other zebrafish lines were conducted according to protocols approved by the Institutional Animal Care and Use Committees in facilities accredited by the Association for Assessment and Accreditation of Laboratory Animal Care International (AAALAC). All zebrafish were fixed overnight at 4°C in 4% paraformaldehyde in phosphate-buffered saline (PBS), except for *Tg(sox17:creER^{T2}; cmlc2:DsRed);Tg(-3.5ubi:loxP-GFP-loxP-mCherry)* zebrafish, which were fixed overnight at 4°C in 4% paraformaldehyde in 0.1 M Pipes, 1 mM $MgSO_4$, 2 mM EGTA, pH 7).

To induce Cre activity and recombination in *Tg(-4.9sox10:creER^{T2});Tg(βactin:loxP-SuperStop-loxP-DsRed)* zebrafish, embryos were dechorionated at 16 hr post-fertilization (hpf) and treated with 5 μM 4-hydroxytamoxifen (4-OHT; Sigma-Aldrich, St. Louis, MO) for 8 hr. As reported in *Mongera et al. (2013)*, 4-OHT treatment of this line for 8 hr from 16 hpf is very effective in yielding Cre-induced recombination in the branchial arches, and was used in *Mongera et al. (2013)* to demonstrate the neural crest origin of gill pillar cells. To induce Cre activity and recombination in *Tg(crestin:creER^{T2});Tg(-3.5ubi:loxP-GFP-loxP-mCherry)* zebrafish, embryos were treated with 20 μM 4-OHT in ethanol at 50% epiboly and again at 24 hpf. To induce Cre activity and recombination in *Tg(sox17:creER^{T2};cmlc2:DsRed);Tg(-3.5ubi:loxP-GFP-loxP-mCherry)* zebrafish, embryos were treated for 3 hr from 5 hpf with 10 μM 4-OHT.

## *Xenopus* neural fold grafts

Experiments using *Xenopus laevis* were conducted in accordance with the UK Animals (Scientific Procedures) Act 1986, with appropriate personal and project licences in place where necessary. Embryos were obtained by in vitro fertilization and initially kept at 14°C in 0.1% modified Barth's saline (MBS). For grafted embryos that would be grown to stage 53 (after the onset of independent feeding), *CMV-GFP* transgenic embryos (*Marsh-Armstrong et al., 1999*) were used as donors and the embryos were grafted and reared at the European *Xenopus* Resource Centre (University of Portsmouth, UK). For grafted embryos that were to be grown to embryonic stages 41–45, GFP-positive donor embryos were made by injecting *cyto-GFP* mRNA into one cell at the two-cell stage, or two cells at the four-cell stage. Briefly, embryos were de-jellied in 2% cysteine and washed several times in 0.1% MBS before being transferred and positioned for injection in a mesh-lined Petri dish filled with 4% Ficoll. Injected embryos were allowed to recover in 4% Ficoll for at least 1 hr before being transferred to 0.1% MBS.

De-jellied GFP-positive embryos and wild-type embryos were allowed to grow to stage 13–17 at 14–18°C. For grafting, embryos were moved to 18 mm Petri dishes lined with plasticine or 1% agarose with depressions and containing a high-salt transplantation solution (1x MBSH: 1x MBS, 0.7 mM CaCl$_2$, 0.02 mM NaCl, supplemented with 2 mM CaCl$_2$ and 2.5 mg/ml gentamycin [Sigma-Aldrich]). The region of the neural folds containing premigratory branchial and vagal neural crest (*Sadaghiani and Thiébaud, 1987*) was removed unilaterally from stage 13–15 wild-type hosts and replaced with GFP-positive tissue from the same region of donor embryos (*Figure 3d*). The grafted tissue was held in place with a small piece of glass coverslip while embryos recovered in transplantation solution for at least 2 hr, before being moved to 0.1% MBS and reared at 18°C. Embryos were overdosed in MS222 (Sigma-Aldrich) in PBS before being fixed in 4% paraformaldehyde in PBS overnight at 4°C.

## Lamprey DiI injections

Experiments using sea lamprey (*Petromyzon marinus*) were conducted according to protocols approved by the California Institute of Technology Institutional Animal Care and Use Committee. Eggs were collected from adults and fertilized as described (*Nikitina et al., 2009*). Embryos were maintained at 18°C in 0.1x or 1x Marc's modified Ringer's (MMR) solution. DiI labeling was performed as described (*Nikitina et al., 2009*) with some modifications. Briefly, E5 embryos (Piavis stages 11–12) were manually dechorionated in 0.1x MMR, then immobilized and oriented in 18-mm Petri dishes that were lined with a fine mesh. Embryos were pressure-injected into the dorsal neural tube using glass capillary tubes filled with 0.5 mg/ml of Cell Tracker-CM-DiI (Invitrogen, Carlsbad, CA) diluted in 0.3 M sucrose (from a 5 mg/ml stock diluted in ethanol). They were allowed to recover for 24 hr, then individually transferred to an uncoated Petri dish containing 1x MMR and allowed to develop to E19 (Piavis stage 17). Embryos were periodically checked and imaged throughout, then fixed in 4% paraformaldehyde in PBS for 1 hr at room temperature.

## Transgenic mouse lines

The following transgenic mouse lines were used: *Wnt1-cre;R26R-YFP* (*Danielian et al., 1998*; *Srinivas et al., 2001*), *Sox17$^{2A-iCre}$;R26$^{R/+}$* (*Engert et al., 2009*; *Soriano, 1999*) and *Sox17$^{2A-iCre}$; R26R$^{tdTomato}$* (*Engert et al., 2009*; *Madisen et al., 2010*). Experiments using these mice were conducted in accordance with the UK Animals (Scientific Procedures) Act 1986, with appropriate personal and project licences in place. Embryos were dissected at appropriate stages and fixed at 4°C overnight in 4% paraformaldehyde in PBS.

## Chicken neural tube grafts

Experiments using chicken (*Gallus gallus domesticus*) embryos were conducted in accordance with the UK Animals (Scientific Procedures) Act 1986, with appropriate personal and project licences in place where necessary. Fertilized wild-type chicken eggs were obtained from Henry Stewart and Co. Ltd., Norfolk, UK. Fertilized GFP-transgenic chicken eggs (*McGrew et al., 2008*) were obtained from the Roslin Institute Transgenic Chicken Facility (Edinburgh, UK), which is funded by the Wellcome Trust and the BBSRC. Fertilized wild-type and GFP-transgenic eggs were incubated in a humidified atmosphere at 38°C for approximately 1.5 days to reach 8–11 somites. The neural tube and associated neural folds between the level of somite one and the caudal end of the seventh somite were dissected from a wild-type host and replaced with the equivalent tissue from a GFP-transgenic donor embryo. At E14, embryos were decapitated and the lungs dissected out and fixed overnight in 4% paraformaldehyde in PBS. At E16.5, embryos were decapitated and fixed overnight in 4% paraformaldehyde in PBS; the lungs were dissected after fixation. The lungs were dehydrated through an ethanol series into 100% ethanol for storage.

## *Xenopus* DiI injections

Stage 14 *Xenopus laevis* embryos were fixed in place in a plasticine dish filled with 1x MBSH supplemented with 2 mM CaCl$_2$, and the endoderm was exposed by cutting a flap into the anterior neural plate (leaving it attached on the anterior side) with tungsten needles, which was folded back to expose the endoderm. A stock solution of 2 mg/ml Cell Tracker-CM-DiI (Invitrogen) in ethanol was diluted 1:10 in 10% sucrose and microinjected into the anterior endoderm (regions 1 and 5 of

*Chalmers and Slack, 2000*) using a glass electrode whose tip was approximately 20 µm in diameter. The endoderm of each embryo was injected at three to five sites. The flap of the neural plate was then folded back in place and pressed down with a small piece of glass coverslip supported on plasticine feet until it healed back in place (approximately 1–2 hr). Embryos were then transferred into 0.1 x MBS containing 25 mg/l gentamicin (Sigma-Aldrich), 400 mg/l penicillin (Sigma-Aldrich) and 400 mg/l streptomycin sulfate (Sigma-Aldrich). At stage 45, tadpoles were overdosed in MS222 (Sigma-Aldrich) in PBS before being fixed in 4% paraformaldehyde in PBS at 4°C overnight for up to several days, then transferred to PBS.

## Ammocoete lamprey
An ammocoete lamprey was euthanized by MS222 (Sigma-Aldrich) overdose, fixed in modified Carnoy's solution (six volumes ethanol: three volumes 37% formaldehyde: 1 volume glacial acetic acid) and dehydrated through an ethanol series into 100% ethanol.

## Embedding and immunostaining
DiI-labeled lamprey embryos, grafted *Xenopus* embryos, some *Tg(-28.5sox10:cre;ef1a:loxP-DsRed-loxP-EGFP)*, *tfap2a^{mob};foxd3^{mos}* zebrafish embryos and their wild-type siblings were dehydrated from PBS into 100% methanol and transferred to 100% isopropanol overnight at 4°C. Embryos were transferred to 1:1 isopropanol:chloroform for 1 hr at 4°C and then to 100% chloroform for 2 hr at −20°C. After warming to room temperature, embryos were transferred to 1:1 chloroform:paraffin wax (Raymond A. Lamb Ltd., Thermo Fisher Scientific, Waltham, MA) at 60°C for 30 min, followed by three 30-min incubations and an overnight incubation at 60°C in paraffin wax. Embryos were embedded in plastic molds and sectioned at 6 µm using a rotary microtome.

DiI-labeled *Xenopus* embryos, *Tg(crestin:creER^{T2});Tg(-3.5ubi:loxP-GFP-loxP-mCherry)*, *Tg(-4.9sox10:creER^{T2});Tg(βactin:loxP-SuperStop-loxP-DsRed)*, some *Tg(-28.5sox10:cre);Tg(ef1a:loxP-DsRed-loxP-EGFP)* and *Tg(sox17:creER^{T2};cmlc2:DsRed);Tg(-3.5ubi:loxP-GFP-loxP-mCherry)* zebrafish in PBS were sucrose-protected before being embedded in 7.5-20% gelatin in plastic molds, flash-frozen in liquid nitrogen and cryosectioned at 6 µm.

Mouse embryos and wild-type zebrafish embryos were sucrose-protected before being embedded in O.C.T. compound (Tissue-Tek, Sakura Finetek, Torrance, CA) in plastic molds, flash-frozen in isopentane on dry ice and cryosectioned at 10–15 µm.

Grafted chicken lungs and the ammocoete lamprey in 100% ethanol were cleared in Histosol (National Diagnostics, Atlanta, GA) and incubated in 1:1 Histosol: paraffin wax (Raymond A. Lamb Ltd.) for 30 min at 60°C, followed by three 30-min incubations and an overnight incubation in paraffin wax at 60°C. They were then embedded in plastic molds and sectioned at 6–10 µm using a rotary microtome.

For immunostaining on paraffin wax sections, slides were de-waxed in Histosol and rehydrated into PBS through a graded ethanol series. Cryosections were allowed to warm to room temperature and washed in PBS. When necessary, gelatin was removed by dipping slides in PBS warmed to 37°C. All anti-serotonin antibodies used required antigen retrieval, which was performed by heating the slides for 30 s in a microwave in 10 mM sodium citrate buffer solution (pH 6), followed by two washes in PBS. Immunostaining was performed as described (*Nikitina et al., 2009*) with slight modifications: slides were incubated overnight at 4°C or at room temperature in primary antibody in blocking solution (10% sheep, goat or donkey serum, as appropriate, in PBS with 0.1% Triton X-100); secondary antibodies were incubated at room temperature for 2 hr or overnight at 4°C. For horse-radish peroxidase detection, slides were incubated in 0.3 mg/ml diaminobenzidine, 0.02% $H_2O_2$, 0.05% Triton X-100 in PBS. After immunostaining, sections were counterstained with the nuclear marker DAPI (1 ng/ml) (Invitrogen) and mounted in Fluoromount G (Southern Biotech, Birmingham, AL).

For whole-mount immunostaining, *Tg(crestin:creER^{T2});Tg(-3.5ubi:loxP-GFP-loxP-mCherry)* zebrafish embryos were incubated for 2 hr in blocking buffer (PBS with 4% bovine serum albumin, 0.3% Triton X-100, 0.02% sodium azide) prior to overnight incubation at 4°C with primary antibodies diluted in blocking buffer. Embryos were washed for 2 hr at room temperature in PBS with 0.3% Triton X-100, then incubated overnight at 4°C in blocking buffer containing secondary antibodies

diluted 1:200 and 1 mg/ml DAPI (Invitrogen) diluted 1:200. After washing for 2 hr in PBS with 0.3% Triton X-100, embryos were suspended in 80% glycerol before mounting.

## Antibodies

Primary antibodies were used against the following antigens: acetylated tubulin [1:250 mouse IgG2b, clone 6-11-B1, T7451 Sigma-Aldrich; previously used in the sea lamprey, e.g., *Barreiro-Iglesias et al. (2008a)*], Ascl1 (Mash1) [1:200 mouse IgG1 (*Lo et al., 1991*), kind gift of F. Guillemot, NIMR, London, UK; 1:100 mouse IgG1, #556604 BD Biosciences, San Jose, CA], DsRed2 (1:100 mouse IgG1, sc-101526 Santa Cruz Biotechnology, Dallas, TX), Elavl3/4 (HuC/D) (1:500 mouse IgG2b, A-21271 Invitrogen), GFP (1:500 rabbit, A-6455 Invitrogen; 1:500 mouse IgG1, #1814460001 Roche, Basel, Switzerland; 1:250 goat, ab6662 Abcam [Cambridge, UK]; 1:150 chicken, ab13970 Abcam), HNK-1 carbohydrate epitope (*Abo and Balch, 1981*; *Voshol et al., 1996*) [for zebrafish neurites (*Metcalfe et al., 1990*): 1:100 mouse IgG1, ZN-12 Developmental Studies Hybridoma Bank; for lamprey neurites (*Barreiro-Iglesias et al., 2008b*): 1:50 mouse IgM, 3H5 Developmental Studies Hybridoma Bank], mCherry (1:250 mouse IgG1, #632543 Clontech Takara Bio USA Inc., Mountain View, CA; 1:200 goat, orb11618 Biorbyt, Cambridge, UK), serotonin (5-hydroxytryptamine, 5-HT) [1:100 (whole-mount) or 1:250 (sections) rabbit, S5545 Sigma-Aldrich, previously used in zebrafish, e.g., *Kuscha et al. (2012)*, bullfrog (*Reyes et al., 2014*) and Arctic lamprey (*Suzuki et al., 2015*); 1:100 rat, MAB352 Merck Millipore, Temecula, CA, previously used in zebrafish (*Sundvik et al., 2013*); 1:250 goat, ab66047 Abcam], Phox2b [1:500 rabbit, kind gift of Jean-François Brunet, Institut de Biologie de l'École Normale Supérieure, Paris, France; previously used in zebrafish (*Coppola et al., 2012*); Tubb3 (neuronal $\beta$-III tubulin) (1:500 mouse IgG2a, clone TUJ1, MMS-435P Covance BioLegend, San Diego, CA), and tyrosine hydroxylase [1:250, rabbit, AB152 Merck Millipore; previously used in zebrafish, e.g., *Yamamoto et al. (2011)*, and sea lamprey, e.g., *Barreiro-Iglesias et al. (2008a)*. (The Developmental Studies Hybridoma Bank was developed under the auspices of the NICHD and is maintained by the University of Iowa, Department of Biological Sciences, Iowa City.) Appropriately matched AlexaFluor or horse-radish peroxidase-conjugated secondary antibodies were obtained from Molecular Probes/Invitrogen.

## Whole-mount in situ hybridization

The lamprey (*P. marinus*) *Phox2* clone (*Häming et al., 2011*) was a kind gift of Marianne Bronner (Caltech, Pasadena, CA, USA). Whole-mount in situ hybridization on lamprey embryos was performed as described (*Nikitina et al., 2009*). After whole-mount in situ hybridization, embryos were incubated in PBS with 5% sucrose for 4 hr at room temperature, followed by incubation overnight at 4°C in 15% sucrose in PBS. Embryos were transferred into pre-warmed 7.5% gelatin in 15% sucrose in PBS and incubated for 1–4 hr at 37°C, then oriented and embedded in molds, frozen by immersion in a dry ice-isopentane solution for 30 s, and cryosectioned at 12–16 µm. Gelatin was removed from the slides by a 5-min incubation in PBS pre-warmed to 37°C.

## In situ hybridization on paraffin wax sections

The chicken *Phox2b* clone (*Stanke et al., 1999*) was a kind gift of Jean-François Brunet (Institut de Biologie de l'École Normale Supérieure, Paris, France). For in situ hybridization on paraffin wax sections, slides were de-waxed in Histosol (National Diagnostics) and rehydrated into diethylpyrocarbonate (DEPC)-treated (Sigma-Aldrich) PBS through a graded ethanol series. In situ hybridization was performed on sections as described (*Miller et al., 2017*).

## X-gal staining

For X-gal staining on cryosections of mouse tissue, the following staining solution was added to slides prior to incubation at 37°C for 2 hr: 5 mM $K_3Fe(CN)_6$, 5 mM $K_4Fe(CN)_6$, 2.7 mM $MgCl_2$ in PBS, supplemented with 75 mg/ml X-gal in dimethyl sulfoxide (DMSO).

## Alcian blue plus hematoxylin and eosin staining

For Alcian blue plus hematoxylin and eosin staining on paraffin sections, slides were de-waxed in Histosol and rehydrated into water through a graded ethanol series. Slides were rinsed in 3% acetic acid before staining in 2 mg/ml Alcian blue (Searle Diagnostic, High Wycombe, UK) in 3% acetic acid

for at least 30 min. Slides were then rinsed in water and treated with 0.3% NaHCO₃, followed by another rinse in running water and staining in Mayer's hematoxylin (Sigma-Aldrich) for 10 min. Slides were stained in 1% aqueous eosin Y solution (BDH) for 8 min, then washed again in running water before dehydration through an ethanol series into 100% ethanol. After washing in Histosol, slides were mounted with DPX (BDH).

## Image capture and processing

Whole-mount images were taken using a Leica MZFLIII microscope (Leica Microsystems, Wetzlar, Germany) fitted with a QImaging MicroPublisher 5.0 RTV camera and QCapture Pro 6.0 software (QImaging, Surrey, BC, Canada); a Zeiss AxioSkop2 microscope fitted with a Zeiss AxioCam HRc camera and Zeiss AxioVision Rel. 4.8 software (Carl Zeiss, Oberkochen, Germany); an Olympus MVX10 microscope (Olympus Corporation, Tokyo, Japan) fitted with a Zeiss AxioCam HRc camera and Zeiss AxioVision Rel. 4.8 software; an Olympus 1 $\times$ 71 inverted microscope fitted with a Hamamatsu ORCA-R2 monochrome camera and HCImage software (Hamamatsu Photonics, Hamamatsu, Japan); a Zeiss LSM 710 confocal microscope with Zeiss ZEN software; and a Zeiss 710 confocal microscope with Zeiss LSM Image Browser (version 4.2.0.121) software, which was used to create three-dimensional images and stack movies. Images of sections were taken using a Zeiss AxioSkop 2 MOT microscope fitted with a QImaging Retiga 2000R camera, a Qimaging RGB pancake and QCapture Pro 6.0 software; a Zeiss Scope.A1 microscope fitted with a Zeiss AxioCam MRm camera and Zeiss ZEN 2012 (blue edition) software; and a Zeiss LSM 780 confocal microscope with Zeiss ZEN 2011 (black edition) software. All images were further processed in Photoshop CS4 (Adobe Systems Inc., San Jose, CA) and/or ImageJ 1.50i software (NIH, Bethesda, MD).

## Statistical analysis

Data analysis and statistical tests were performed using Microsoft Excel and GraphPad Prism 7 software (GraphPad Software, Inc., La Jolla, CA). Data sets were tested for normality using the Shapiro-Wilk test (alpha = 0.05) and for equality of variance using an F test (p=0.38), and compared using an unpaired two-tailed Student's t-test. Data are presented as mean $\pm$ standard deviation (s. d.).

## Acknowledgements

This work was funded by the Wellcome Trust (Ph.D. Studentship 086804/Z/08/Z to DH; Senior Investigator Award 102889/Z/13/Z to AST), the NIDCR/NIH (R21-DE021509 to SF; R01-DE018477 to EWK), the NIDDK/NIH (1DP2DK098092 to PDSD), the NIH (R01-HL092217 to EWK), the Zebrafish Initiative of the Vanderbilt University Academic Venture Capital Fund (to EWK), the Vanderbilt International Scholar Program (to GU), the HFSP (Long-Term Fellowship to CM) and the Swiss National Science Foundation (Advanced Postdoctoral Fellowship and Professorship to CM). For financial support, HL would like to thank the Helmholtz Society (Helmholtz Portfolio Theme 'Metabolic Dysfunction and Common Disease'), the Helmholtz Alliance (Imaging and Curing Environmental Metabolic Disease), and the German Center for Diabetes Research (DZD e.V.). Additional support for DH was provided by the Cambridge Trusts, the Cambridge Philosophical Society, the Oppenheimer Memorial Trust and Trinity College Oxford. DH thanks Tatjana Sauka-Spengler for her support and advice. AM thanks Christiane Nüsslein-Volhard for her support and advice. Thanks to Colin Sharpe, Matt Guille and Alan Jafkins (University of Portsmouth, Portsmouth, UK) for access to transgenic *Xenopus* embryos and rearing of grafted embryos at the European *Xenopus* Research Centre (funded by the Wellcome Trust, the BBSRC and NC3Rs); to Christine Holt and Vasja Urbančič (University of Cambridge, Cambridge, UK) for provision of wild-type *Xenopus* embryos; to Marianne Bronner (California Institute of Technology, Pasadena, CA, USA) and David Parker (University of Cambridge, Cambridge, UK) for access to lampreys, and to Helen Sang and Adrian Sherman (Roslin Institute, Edinburgh, UK) for providing transgenic chicken eggs (Roslin Institute Transgenic Chicken Facility, Edinburgh, UK, funded by the Wellcome Trust and the BBSRC). Thanks to Perrine Barraud (University of Cambridge) for synthesizing the chicken *Phox2b* riboprobe, to Jana Koth (University of Oxford) for assistance in identifying zebrafish blood vessels, and to Giacomo Zaccone (University of Messina, Messina, Italy) for comments on an earlier version of the manuscript.

# Additional information

## Funding

| Funder | Grant reference number | Author |
|---|---|---|
| Wellcome | 086804/Z/08/Z | Dorit Hockman |
| National Institute of Dental and Craniofacial Research | R21-DE021509 | Shannon Fisher |
| Zebrafish Initiative of the Vanderbilt University Venture Capital Fund | | Ela W Knapik |
| Vanderbilt International Scholar Program | Graduate Student Scholarship | Gokhan Unlu |
| Swiss National Science Foundation | Advanced Postdoctoral Fellowship and Professorship | Christian Mosimann |
| Cambridge Trusts | Graduate Student Scholarship | Dorit Hockman |
| Cambridge Philosophical Society | Graduate Student Scholarship | Dorit Hockman |
| Oppenheimer Memorial Trust | Graduate Student Scholarship | Dorit Hockman |
| Trinity College Oxford | Junior Research Fellowship | Dorit Hockman |
| Wellcome | 102889/Z/13/Z | Abigail S Tucker |
| National Institute of Dental and Craniofacial Research | R01-DE018477 | Ela W Knapik |
| National Institute of Diabetes and Digestive and Kidney Diseases | 1DP2DK098092 | Duc S Dong |
| Human Frontier Science Program | Long-Term Fellowship | Christian Mosimann |
| Helmholtz-Gemeinschaft | Helmholtz Portfolio Theme 'Metabolic Dysfunction and Common Disease' | Heiko Lickert |
| Helmoltz Alliance | Imaging and Curing Environmental Metabolic Disease | Heiko Lickert |
| German Center for Diabetes Research | | Heiko Lickert |
| National Institutes of Health | R01-HL092217 | Ela W Knapik |

The funders had no role in study design, data collection and interpretation, or the decision to submit the work for publication.

## Author contributions

DH, conceived and designed the experiments with CVHB, performed all experiments except the *Xenopus* DiI injections reported in Figure 5e-i' (GS), the chicken neural fold grafts reported in Figure 5-figure supplement 1c-e' (AJB), the whole-mount zebrafish immunostaining reported in Figure 5-figure supplement 2 (KPG) and the chicken Phox2b immunostaining reported in Figure 6m,m' (BJ), analyzed all the results except the zebrafish skin NEC data (KPG), prepared all the figures and wrote the manuscript with CVHB; AJB, performed the chicken neural fold grafts reported in Figure 5-figure supplement 1c-e', provided *Wnt1:cre;R26R:YFP* mouse embryos and commented on the manuscript; GS, performed the *Xenopus* DiI injections reported in Figure 5e-i' and commented on the manuscript; KPG, performed, imaged and analyzed the whole-mount zebrafish immunostaining data reported in Figure 5-figure supplement 2, provided (with JJL and PDSD) 4-OHT-treated *Tg(sox17:creER^{T2};cmlc2:DsRed);Tg(-3.5ubi:loxP-GFP-loxP-mCherry)* zebrafish larvae, and commented on the

manuscript; BJ, performed and imaged the chicken Phox2b immunostaining reported in Figure 6m, m', and commented on the manuscript; AM, provided 4-OHT-treated *Tg(-4.9sox10:creER^{T2});Tg(βactin:loxP-SuperStop-loxP-DsRed)* zebrafish larvae and commented on the manuscript; SF, provided *Tg(-28.5sox10:cre);Tg(ef1a:loxP-DsRed-loxP-EGFP)* zebrafish larvae and commented on the manuscript; GU, provided (with EWK) *tfap2a^{mob};foxd3^{mos}* zebrafish larvae and commented on the manuscript; EWK, provided (with GU) *tfap2a^{mob};foxd3^{mos}* zebrafish larvae and commented on the manuscript; CKK, provided (with CM and LIZ) 4-OHT-treated *Tg(crestin:creER^{T2});Tg(-3.5ubi:loxP-GFP-loxP-mCherry)* zebrafish larvae and juveniles, and commented on the manuscript; CM, provided (with CKK and LIZ) 4-OHT-treated *Tg(crestin:creER^{T2});Tg(-3.5ubi:loxP-GFP-loxP-mCherry)* zebrafish larvae and juveniles, and commented on the manuscript; LIZ, provided (with CKK and CM) 4-OHT-treated *Tg(crestin:creER^{T2});Tg(-3.5ubi:loxP-GFP-loxP-mCherry)* zebrafish larvae and juveniles, and commented on the manuscript; JJL, provided (with KPG and PDSD) 4-OHT-treated *Tg(sox17:creER^{T2};cmlc2:DsRed);Tg(-3.5ubi:loxP-GFP-loxP-mCherry)* zebrafish larvae, and commented on the manuscript; PDSD, provided (with KPG and JJL) 4-OHT-treated *Tg(sox17:creER^{T2};cmlc2:DsRed);Tg(-3.5ubi:loxP-GFP-loxP-mCherry)* zebrafish larvae, and commented on the manuscript; HL, provided access to the *Sox17^{2A-iCre}* mouse line and commented on the manuscript; AST, provided *Sox17^{2A-iCre/+};R26^{R/+}* mouse embryos and cryosections of *Sox17^{2A-iCre/+};R26R^{tdTomato}* mouse embryos, and commented on the manuscript; CVHB, conceived and designed the experiments with DH and wrote the manuscript with DH

### Author ORCIDs
Leonard I Zon, http://orcid.org/0000-0003-0860-926X
Clare V H Baker, http://orcid.org/0000-0002-4434-3107

### Ethics
Animal experimentation: Experiments using *Tg(-4.9sox10:creER^{T2});Tg(βactin:loxP-SuperStop-loxP-DsRed)* zebrafish were conducted in compliance with the regulations of the Regierungspräsidium Tübingen and the Max Planck Society. Experiments using all other zebrafish lines were conducted according to protocols approved by the Institutional Animal Care and Use Committees in facilities accredited by the Association for Assessment and Accreditation of Laboratory Animal Care International (AAALAC). Experiments using *Xenopus laevis* were conducted in accordance with the UK Animals (Scientific Procedures) Act 1986, with appropriate personal and project licences in place where necessary. Experiments using sea lamprey (*Petromyzon marinus*) were conducted according to protocols approved by the California Institute of Technology Institutional Animal Care and Use Committee. Experiments using transgenic mice were conducted in accordance with the UK Animals (Scientific Procedures) Act 1986, with appropriate personal and project licences in place. Experiments using chicken (*Gallus gallus domesticus*) embryos were conducted in accordance with the UK Animals (Scientific Procedures) Act 1986, with appropriate personal and project licences in place where necessary.

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
