## [Decision Letter]

Thank you for submitting your article "Evolution of the hypoxia-sensitive cells involved in amniote respiratory reflexes" for consideration by *eLife*. Your article has been reviewed by three peer reviewers, and the evaluation has been overseen by Robb Krumlauf as the Senior Editor and Reviewing Editor. The reviewers have opted to remain anonymous.

The reviewers have discussed the reviews with one another and the Senior Editor has drafted this decision to help you prepare a revised submission.

Summary:

This manuscript focuses on the evolutionary origins of hypoxia-sensitive cells in the transition from anamniotes to amniotes. The work focuses largely on lineage and physical fate mapping studies to trace the embryological origin of the neuroendocrine cells (NEC) in gills of non-amniotic species (zebrafish, lamprey and frogs) and to test whether these NEC correspond to the glomus or PNEC of amniotes (mice). They show that gill NEC do not derive from neural crest cells, unlike the glomus cells, but arise by in situ differentiation of the gill epithelium and thus are rather homologous to amniote pulmonary NEC (PNEC) than to glomus NEC. The study addresses a complex question and includes various genetic and cell labeling approaches in zebrafish, *Xenopus*, lamprey and mouse. This is a large body of work that is both interesting and carefully done. However in the present version several issues are left to be resolved. Many can be handled by modifications to the text, softening statements and providing more details or clarity. However, there is also a need for additional experimental data.

Major revisions:

1) Results from the lineage tracing and knockout studies provided here are overall clear and supportive in their claim that PNECs in amniotes and NEC anamniotes are endoderm-derived. However, there are important gaps of knowledge and uncertainties regarding the role of NEC as hypoxia-sensitive cells in the species they analyze. This raises questions on whether the title of this paper "Evolution of the hypoxia-sensitive cells in amniote respiratory reflexes" can be justified. The authors need to deal with these issues in the text and consider modifying the title. Specifically:

A) Patch-clamp studies provide evidence that gill NECs of zebrafish and catfish are hypoxia sensitive. However, there is a lack of evidence that the oropharyngeal putative NEC (innervated serotonergic cells) of anamniotes are hypoxia-sensitive (subsection “NECs are endoderm-derived, suggesting homology with PNECs, not glomus cells”, first paragraph). This weakens their proposed hypothesis that "hypoxia-sensitive NECs in the epithelia of the gills and orobranchial cavity of ancestral vertebrates were retained in air-breathing organs of both anamniotes and amniotes".

B) The authors state that putative hypoxia-responsive NECs have been reported in the skin of anamniotes (zebrafish, killifish). However, they do not comment on whether these NEC differentiate in situ from ectodermal skin progenitors or originate from migrating neural crest cells (like melanocytes). This may add complexity to their model, given that oropharyngeal NEC (shown to be non-neural crest-derived) have not yet been proven to be hypoxia-sensitive. Are there hypoxia-responsive and unresponsive NEC in anamniotes?

2) Points related to the issue of homology:

A) What does define structures as "homologous"? Given the focus of this manuscript, the authors should provide a definition or give more stringent criteria to consider structures "homologous".

B) What is the functional significance of the homology between these cells in anamniotes and amniotes? Does this provide any evolutionary advantage? The discussion does not lead to clear insights into this issue and it should be addressed.

C) In the third paragraph of the Introduction: The authors mention that "if glomus cells and NECs are homologous, they should share a common embryonic origin" and in the first paragraph of the subsection “NECs are endoderm-derived, like PNECs”: “Overall our data show that anamniote NECs are not neural crest-derived, hence cannot be homologous to glomus cells". The authors need to properly qualify these statements. While embryonic origins generally do support homology (defined as similarity resulting from common ancestry), especially for more complex tissues and structures, modern EvoDevo has shown repeatedly that homologous cell types (cells whose similarly is so extensive that they must be evolutionarily related) can sometimes have different embryonic origins. Perhaps the most dramatic example of this are ectomesenchymal NCC derivatives, e.g. NCC-derived smooth muscle and bone are clearly homologous to mesoderm-derived smooth muscle or bone, despite originating from different germ layers. If NECs and Glomus cells were shown to activate the same genetic network, they could be considered homologous despite disparate embryonic origins.

D) With respect to the anamniote evolutionary homologs of glomus cells, the authors only establish a superficial similarity. There is no proof that the catecholaminergic neural crest-derived cells in zebrafish have any hypoxia-sensing role, the hypothesis for homology is highly speculative. The authors need to qualify or soften their hypothesis by noting this lack of evidence for hypoxia-sensing.

3) Aside from serotonin immunoreactivity, chromium staining/tyrosine hydroxylase, and various general neural markers, nothing is known about the gene-regulatory identity of NECs, PNECs, glomus cells, and the newly described chromaffin cells. Thus, it is formally possible that glomus cells and NECs may activate identical gene-regulatory programs, while PNECs and NECs could be very different at the level of the transcription factors and other developmental regulators they utilize, and are thus similar due to convergence. The use of serotonin (5-HT) as a marker of NECs should be confirmed with a second NEC-specific marker that does not label neurons as well. There is evidence for serotonin-negative NECs (Jonz et al. 2004); it should be mentioned that the NECs described in this study are not comparable to those. Ideally, it would be important to have 1 or more molecular markers for developmental regulators (aside from terminal differentiation/effector proteins like serotonin) that confirm the proposed homologies.

4) Although figures are well done, the overall choice of colors used in immunofluorescence (IF) images (green/magenta) does not yield optimal assessment of the patterns or resolution details. This is particularly problematic when double labeling needs to be demonstrated. In several instances there seems to be a disconnect between the perceived and the described co-localization of signals in the figures and text. A combination of red and green should minimize this issue and is recommended to be used in all IF images.

5) In Figure 5—figure supplement 1 there are at least 2 clearly labeled Wnt1-Cre;R26-YFP+ epithelial cells in the field. Both are closely associated with Ascl1-expressing cells (or could even be expressing Ascl1, depending on sectioning angle). These cells are not indicated in the figure or commented on in the text. As noted above, the use of red/green colors in these images would likely help in clarifying whether there is overlap in YFP-Ascl1. How frequently these cells are seen? Do they appear at other developmental stages and how frequently? Is this an artifact or real? This is relevant, given the focus of this paper on the contribution of neural crest cells to the NEC lineage. In addition, neural crest-derived cells in the epithelium, if confirmed, is an important observation.

6) It is interesting to see that the number of putative NECs seems to be unchanged in zebrafish that are missing neural crest derivatives. Their development seems to be unaffected even when their usual developmental environment is largely disrupted. This is an intriguing result and would be made more so by examination of a greater number of embryos as the current statistics show a lot of variation.

7) The lineage tracing of Sox17 using reporter mice in Figure 5 does not include double staining (βGal-5HT). Moreover, it is not explained that this reporter labels both epithelial and mesenchymal components nor why it was used. Half of 5-HT positive cells were also positive for a sox17 induced reporter in zebrafish (Figure 5). Is this a consequence of the sections including many non-NEC (or otherwise non-endoderm derived) serotonergic cells, or does it reflect somewhat low levels of the inducible Cre in these fish, leading to unsaturated reporter switching? Some discussion of this in the text would be appreciated.

---

## [Author Response]

*Major revisions:*

*1) Results from the lineage tracing and knockout studies provided here are overall clear and supportive in their claim that PNECs in amniotes and NEC anamniotes are endoderm-derived. However, there are important gaps of knowledge and uncertainties regarding the role of NEC as hypoxia-sensitive cells in the species they analyze. This raises questions on whether the title of this paper "Evolution of the hypoxia-sensitive cells in amniote respiratory reflexes" can be justified. The authors need to deal with these issues in the text and consider modifying the title. Specifically:*

*A) Patch-clamp studies provide evidence that gill NECs of zebrafish and catfish are hypoxia sensitive. However, there is a lack of evidence that the oropharyngeal putative NEC (innervated serotonergic cells) of anamniotes are hypoxia-sensitive (subsection “NECs are endoderm-derived, suggesting homology with PNECs, not glomus cells”, first paragraph). This weakens their proposed hypothesis that "hypoxia-sensitive NECs in the epithelia of the gills and orobranchial cavity of ancestral vertebrates were retained in air-breathing organs of both anamniotes and amniotes".*

Indeed, there is no direct evidence that the putative NECs in the orobranchial epithelium are hypoxia-sensitive, although we did note in the Introduction that: “Denervation experiments in various fishes (a shark, as well as some teleosts) have shown that hypoxia-sensitive and/or CO_2_sensitive chemoreceptors involved in ventilation responses are located in the orobranchial cavity, as well as in the gills, although their cellular identity has not been confirmed (reviewed by Milsom, 2012).” We have now moved the above text and associated information to the Discussion (subsection “Gill NECs are endoderm-derived, suggesting shared ancestry with PNECs, not glomus cells”, first paragraph), where its relevance will hopefully be clearer. We have also clarified that our hypothesis for PNEC evolution is speculative, and that testing this hypothesis must await evidence for the hypoxia-sensitivity of the putative NECs in the orobranchial epithelium of anamniotes (see aforementioned paragraph). We have also amended the “Hypoxia-sensitive cells” label in the schematic summary figure (Figure 8, formerly Figure 7) to “Putative hypoxia-sensitive cells”.

*B) The authors state that putative hypoxia-responsive NECs have been reported in the skin of anamniotes (zebrafish, killifish). However, they do not comment on whether these NEC differentiate in situ from ectodermal skin progenitors or originate from migrating neural crest cells (like melanocytes). This may add complexity to their model, given that oropharyngeal NEC (shown to be non-neural crest-derived) have not yet been proven to be hypoxia-sensitive. Are there hypoxia-responsive and unresponsive NEC in anamniotes?*

The embryonic origin of putative skin NECs (identified so far only in embryonic zebrafish and adult killifish, i.e., two teleost fish species) had not previously been investigated. In response to this comment, we performed additional neural crest lineage-tracing experiments in zebrafish: we now provide experimental data showing that the putative skin NECs in embryonic zebrafish are not neural crest-derived (Figure 2—figure supplement 1; Video 1 and Video 2; subsection “Zebrafish NECs are not neural crest-derived”, fourth paragraph).

The comment also made us realise that our original wording about the putative skin NECs was ambiguous: they have not been shown directly (e.g. by patch-clamp) to be hypoxia-sensitive, although the putative NECs increase in area in response to hypoxia in killifish, while the usual decline in the number of the putative zebrafish NECs seen with age is reduced or delayed by hypoxia, and accelerated by hyperoxia. We have clarified this in the text (see aforementioned paragraph and subsection “Gill NECs are endoderm-derived, suggesting shared ancestry with PNECs, not glomus cells”, second paragraph).

Overall, we feel that our title is justified. Zebrafish gill NECs, mammalian carotid body glomus cells and mammalian PNECs have all been shown to be directly sensitive to hypoxia, and are all proposed to play a role in mediating respiratory reflexes. By exploring their development, and suggesting a new model for the evolution of both glomus cells and PNECs in light of our experimental data, our manuscript explores the evolution of the hypoxiasensitive cells involved in amniote respiratory reflexes. Direct physiological studies are of course still needed to confirm the hypoxia-sensitivity of the putative NECs in the orobranchial epithelium, as well as the putative gill and orobranchial NECs in *Xenopus* and lamprey: we hope that our manuscript will stimulate further work in this area. (If you insist, we can of course change the title, but we would appreciate suggestions as to what you feel would be more appropriate.)

*2) Points related to the issue of homology:*

*A) What does define structures as "homologous"? Given the focus of this manuscript, the authors should provide a definition or give more stringent criteria to consider structures "homologous".*

We now specify that we are testing whether gill NECs and glomus cells derive from a common ancestral cell population (Introduction, third and fourth paragraphs; subsection “Gill NECs are endoderm-derived, like PNECs”, first paragraph; Discussion subsection “Gill NECs are endoderm-derived, suggesting shared ancestry with PNECs, not glomus cells”, heading and first paragraph).

*B) What is the functional significance of the homology between these cells in anamniotes and amniotes? Does this provide any evolutionary advantage? The discussion does not lead to clear insights into this issue and it should be addressed.*

We have added text to the first paragraph of the Discussion to address this comment.

*C) In the third paragraph of the Introduction: The authors mention that "if glomus cells and NECs are homologous, they should share a common embryonic origin" and in the first paragraph of the subsection “NECs are endoderm-derived, like PNECs”: “Overall our data show that anamniote NECs are not neural crest-derived, hence cannot be homologous to glomus cells". The authors need to properly qualify these statements. While embryonic origins generally do support homology (defined as similarity resulting from common ancestry), especially for more complex tissues and structures, modern EvoDevo has shown repeatedly that homologous cell types (cells whose similarly is so extensive that they must be evolutionarily related) can sometimes have different embryonic origins. Perhaps the most dramatic example of this are ectomesenchymal NCC derivatives, e.g. NCC-derived smooth muscle and bone are clearly homologous to mesoderm-derived smooth muscle or bone, despite originating from different germ layers. If NECs and Glomus cells were shown to activate the same genetic network, they could be considered homologous despite disparate embryonic origins.*

This is a very valid point brought up by the reviewers: separate embryonic origin does not necessarily rule out homology. As suggested, we investigated whether NECs and glomus cells activate the same genetic network. We have added new expression data to show that Phox2b, which is essential for glomus cell development (Dauger et al., 2003, Development, 130, 663542), is not expressed by zebrafish gill NECs, or by their putative counterparts in lamprey, or by putative NECs in the orobranchial epithelium (Figure 6’ in the revised manuscript; subsection “NECs do not express Phox2b, which is essential for glomus cell development”, first paragraph). Hence, NECs and glomus cells cannot be considered to be homologous cell types by virtue of activating the same genetic network. We now refer to this possibility explicitly (Introduction, last paragraph, subsection “NECs do not express Phox2b, which is essential for glomus cell development”, first and last paragraphs, subsection “Gill NECs are endoderm-derived, suggesting homology shared ancestry with PNECs, not glomus cells”, last paragraph). We also show that, like NECs, PNECs in mouse and chicken do not express Phox2b (Figure 6’ in the revised manuscript; subsection “NECs do not express Phox2b, which is essential for glomus cell development”, second paragraph).

*D) With respect to the anamniote evolutionary homologs of glomus cells, the authors only establish a superficial similarity. There is no proof that the catecholaminergic neural crest-derived cells in zebrafish have any hypoxia-sensing role, the hypothesis for homology is highly speculative. The authors need to qualify or soften their hypothesis by noting this lack of evidence for hypoxia-sensing.*

We agree that this is a speculative hypothesis: we have qualified the wording to indicate this (Introduction, last paragraph, subsection “Candidate glomus cell homologues in fish: neural crest-derived chromaffin cells associated with pharyngeal arch blood vessels”, last paragraph, subsection “A new hypothesis for carotid body evolution”, second paragraph and subsection “A new model for amniote hypoxia-sensitive cell evolution”), and added sentences noting the need for evidence of hypoxia-sensitivity (see the aforementioned paragraphs). We have also amended the “Hypoxia-sensitive cells” label in the schematic summary figure (Figure 8, formerly Figure 7) to “Putative hypoxia-sensitive cells”.

*3) Aside from serotonin immunoreactivity, chromium staining/tyrosine hydroxylase, and various general neural markers, nothing is known about the gene-regulatory identity of NECs, PNECs, glomus cells, and the newly described chromaffin cells. Thus, it is formally possible that glomus cells and NECs may activate identical gene-regulatory programs, while PNECs and NECs could be very different at the level of the transcription factors and other developmental regulators they utilize, and are thus similar due to convergence.*

As noted in the response to point 2C, we have now provided data showing that NECs and PNECs do not express Phox2b, hence are genetically distinct from glomus cells.

*The use of serotonin (5-HT) as a marker of NECs should be confirmed with a second NEC-specific marker that does not label neurons as well.*

Unfortunately, there are no NEC-specific markers. Serotonin is the conventional marker for NECs. Other published NEC markers, such as the synaptic vesicle protein SV2, label neurons as well. We have added more information into the Introduction about how fish gill NECs were first identified, and the use of serotonin as the conventional marker for these cells (Introduction, second paragraph).

*There is evidence for serotonin-negative NECs (Jonz et al. 2004); it should be mentioned that the NECs described in this study are not comparable to those.*

We now include information about these cells in the Introduction (second paragraph). We note that both Jonz et al. (2004, J. Physiol., 560, 737-52) and the previous paper by the same group reporting the existence of serotonin-negative NECs (using expression of the synaptic vesicle protein SV2; Jonz & Nurse, 2003, J. Comp. Neurol., 461, 117) speculate that these cells are immature NECs; we have included this in our description (see aforementioned paragraph).

*Ideally, it would be important to have 1 or more molecular markers for developmental regulators (aside from terminal differentiation/effector proteins like serotonin) that confirm the proposed homologies.*

As noted in the response to point 2C, we have added new data showing that gill NECs, putative gill NECs, putative orobranchial NECs, and PNECs, all lack Phox2b expression, which is essential for glomus cell development (Dauger et al., 2003, Development, 130, 6635-6642).

*4) Although figures are well done, the overall choice of colors used in immunofluorescence (IF) images (green/magenta) does not yield optimal assessment of the patterns or resolution details. This is particularly problematic when double labeling needs to be demonstrated. In several instances there seems to be a disconnect between the perceived and the described co-localization of signals in the figures and text. A combination of red and green should minimize this issue and is recommended to be used in all IF images.*

The use of red and green together disadvantages red-green colour-blind readers, i.e., 5-10% of all male readers. Using green with magenta, instead of red, is recommended to enable colour-blind readers to see the data being presented – for further information, please see the ‘Color Universal Design’ website: http://jfly.iam.u-tokyo.ac.jp/color/index.html.

We do not know which specific images are deemed to be problematic. Cells of interest are indicated with arrows and single-channel images are shown, when necessary, to confirm localization for any channel signal that may be harder to see in the merged image. However, we have amended some figure panels (Figure 1—figure supplement 1; Figure 5 panels E-E’’’) to show the single channels in grey-scale and the triple-merge image in red, green and blue, since the Color Universal Design website says that this approach is acceptable for colour-blind readers.

*5) In Figure 5—figure supplement 1 there are at least 2 clearly labeled Wnt1-Cre;R26-YFP+ epithelial cells in the field. Both are closely associated with Ascl1-expressing cells (or could even be expressing Ascl1, depending on sectioning angle). These cells are not indicated in the figure or commented on in the text. As noted above, the use of red/green colors in these images would likely help in clarifying whether there is overlap in YFP-Ascl1. How frequently these cells are seen? Do they appear at other developmental stages and how frequently? Is this an artifact or real? This is relevant, given the focus of this paper on the contribution of neural crest cells to the NEC lineage. In addition, neural crest-derived cells in the epithelium, if confirmed, is an important observation.*

The two fainter green, out-of-focus spots in Figure 5—figure supplement 1 panels A,A’ are an artefact from the GFP immunostaining (we were unable to find any epithelial staining in panels B,B’). We have replaced panels A,A’ with a higher-power image of the same section, to try to show this more clearly, and have commented on the fainter out-of-focus spots in the legend. We did not see any reporter-positive cells in the epithelium, consistent with an earlier, comprehensive analysis of the neural crest contribution to the intrinsic ganglia of the lungs using the same *Wnt1-Cre/R26R-EYFP* mouse line (Freem et al., 2010, J. Anat., 217, 651-664).

Similarly, the recent analysis of *Wnt1-Cre;;Rosa26^Zsgreen/+^*embryos showing that there was no neural crest contribution to PNECs – which we confirm in our manuscript – described reporter expression in the lungs only in the subepithelial intrinsic ganglia (Kuo & Krasnow, 2015, Cell, 163, 394-405).

*6) It is interesting to see that the number of putative NECs seems to be unchanged in zebrafish that are missing neural crest derivatives. Their development seems to be unaffected even when their usual developmental environment is largely disrupted. This is an intriguing result and would be made more so by examination of a greater number of embryos as the current statistics show a lot of variation.*

We have shown that the putative NECs in the orobranchial epithelium differentiate in situ. There is no a priori reason to think that the development of the orobranchial epithelium would be disrupted by the absence of neural crest cells in the surrounding mesenchyme: neural crest cells are not required for pharyngeal pouch formation, for example (Veitch et al., 1999, Curr. Biol., 9, 1481-1484). Adding more numbers for these experiments, which involve counting all putative NECs in the entire orobranchial region, would not be trivial, and we do not feel that this would be relevant for the main thrust of the paper. We have added a sentence to the main text to clarify that we counted all putative NECs in the orobranchial region, with 294 putative NECs counted in total across three *tfap2a^mob^;;foxd3^mos^* larvae and 244 putative NECs counted in total across three wild-type siblings (subsection “Zebrafish NECs are not neural crest-derived”, third paragraph). Also in this paragraph, we realised that we had not cited the original papers describing the *tfap2a^mob^* and *foxd3^mos^* mutants: now added.

*7) The lineage tracing of Sox17 using reporter mice in Figure 5 does not include double staining (βGal-5HT).*

We have performed additional experiments using the *Sox17^2A-iCre^*driver line crossed to the *R26R^tdTomato^*reporter line, which has enabled double immunostaining (Figure 5).

*Moreover, it is not explained that this reporter labels both epithelial and mesenchymal components nor why it was used.*

We explained in the legend to Figure 5 that in *Sox17^2A-iCre/+^;R26^R/+^*mice, all endoderm-derived lineages, as well as vascular endothelial cells and the haematopoietic system, constitutively express β-galactosidase. We have now added this information about the *Sox17^2AiCre^*driver line into the main text (subsection “Gill NECs are endoderm-derived, like PNECs”, first paragraph).

*Half of 5-HT positive cells were also positive for a sox17 induced reporter in zebrafish (Figure 5). Is this a consequence of the sections including many non-NEC (or otherwise non-endoderm derived) serotonergic cells, or does it reflect somewhat low levels of the inducible Cre in these fish, leading to unsaturated reporter switching? Some discussion of this in the text would be appreciated.*

The relatively low labelling efficiency likely results from a lack of optimization of the 4-OHT dose for the *Tg(sox17:creER^T2^;cmlc2:DsRed*) driver in combination with this particular switchable reporter line (Mosimann et al., 2011). We now state this in the text (subsection “Gill NECs are endoderm-derived, like PNECs”, third paragraph).